# Probing the catalytic functions of Bub1 kinase using the small molecule inhibitors BAY-320 and BAY-524

Anna P Baron[1†], Conrad von Schubert[1†], Fabien Cubizolles[1], Gerhard Siemeister[2], Marion Hitchcock[2], Anne Mengel[2], Jens Schröder[2], Amaury Fernández-Montalván[2], Franz von Nussbaum[2], Dominik Mumberg[2], Erich A Nigg[1*]

[1]Biozentrum, University of Basel, Basel, Switzerland; [2]Global Drug Discovery, Bayer Pharma AG, Berlin, Germany

**Abstract** The kinase Bub1 functions in the spindle assembly checkpoint (SAC) and in chromosome congression, but the role of its catalytic activity remains controversial. Here, we use two novel Bub1 inhibitors, BAY-320 and BAY-524, to demonstrate potent Bub1 kinase inhibition both in vitro and in intact cells. Then, we compared the cellular phenotypes of Bub1 kinase inhibition in HeLa and RPE1 cells with those of protein depletion, indicative of catalytic or scaffolding functions, respectively. Bub1 inhibition affected chromosome association of Shugoshin and the chromosomal passenger complex (CPC), without abolishing global Aurora B function. Consequently, inhibition of Bub1 kinase impaired chromosome arm resolution but exerted only minor effects on mitotic progression or SAC function. Importantly, BAY-320 and BAY-524 treatment sensitized cells to low doses of Paclitaxel, impairing both chromosome segregation and cell proliferation. These findings are relevant to our understanding of Bub1 kinase function and the prospects of targeting Bub1 for therapeutic applications.

*For correspondence: erich.
nigg@unibas.ch

[†]These authors contributed
equally to this work

Competing interest: See
page 21

Reviewing editor: Jon Pines,
The Gurdon Institute, United
Kingdom

## Introduction

During eukaryotic cell division, the spindle assembly checkpoint (SAC) contributes to ensure the accuracy of chromosome segregation. This evolutionarily conserved surveillance mechanism monitors the status of kinetochore (KT)-microtubule (MT) interactions and delays anaphase onset until all chromosomes have undergone bipolar attachment to the spindle. Both KT-MT interactions and SAC activity are regulated by several KT-associated protein kinases, including Aurora B, Monopolar spindle 1 (Mps1) and Budding uninhibited by benzimidazoles 1 (Bub1) (*Maciejowski et al., 2010*; *Meraldi and Sorger, 2005*; *Santaguida et al., 2011*). SAC activity depends on a diffusible inhibitor of the ubiquitin ligase anaphase-promoting complex/cyclosome (APC/C), termed mitotic checkpoint complex (MCC) (*Foley and Kapoor, 2013*; *Lara-Gonzalez et al., 2012*; *Musacchio, 2011*; *Sacristan and Kops, 2015*). Once the SAC is silenced in response to chromosome biorientation, APC/C activation then triggers the onset of chromatid separation and mitotic exit, respectively (*Funabiki and Wynne, 2013*).

The serine/threonine kinase Bub1 is one of the first proteins to accumulate at unattached kinetochores (*Jablonski et al., 1998*). Its recruitment is governed by Mps1-dependent phosphorylation of MELT motifs on the KMN complex member KNL-1 (*London et al., 2012*; *Overlack et al., 2015*; *Shepperd et al., 2012*; *Vleugel et al., 2013*; *Yamagishi et al., 2012*). Bub1 has been implicated in the regulation of chromosome cohesion, KT-MT interactions and SAC function. In particular, Bub1 was shown to be important for the centromere/KT recruitment of Shugoshin proteins (Sgo1 and

**eLife digest** The DNA in our cells is packaged into structures called chromosomes. When a cell divides, these chromosomes need to be copied and then correctly separated so that both daughter cells have a full set of genetic information. Errors in separating chromosomes can lead to the death of cells, birth defects or contribute to the development of cancer.

Chromosomes are separated by an array of protein fibers called the mitotic spindle. A surveillance mechanism known as the spindle assembly checkpoint prevents the cell from dividing until all the chromosomes have properly attached to the spindle. A protein called Bub1 is a central element of the SAC. However, it was not clear whether Bub1 works primarily as an enzyme or as a scaffolding protein.

Baron, von Schubert et al. characterized two new molecules that inhibit Bub1's enzyme activity and used them to investigate what role the enzyme plays in the spindle assembly checkpoint in human cells. The experiments compared the effects of these inhibitors to the effects of other molecules that block the production of Bub1. Baron, von Schubert et al.'s findings suggest that Bub1 works primarily as a scaffolding protein, but that the enzyme activity is required for optimal performance.

Further experiments show that when the molecules that inhibit the Bub1 enzyme are combined with paclitaxel – a widely used therapeutic drug – cancer cells have more difficulties in separating their chromosomes and divide less often. The new inhibitors used by Baron, von Schubert et al. will be useful for future studies of this protein in different situations. Furthermore, these molecules may have the potential to be used as anti-cancer therapies in combination with other drugs.

Sgo2), the chromosomal passenger complex (CPC) comprising Aurora B kinase, CENP-E, CENP-F, BubR1, Mad1 and Mad2 (*Boyarchuk et al., 2007*; *Kitajima et al., 2005*; *Klebig et al., 2009*; *Liu et al., 2013*; *Perera et al., 2007*; *Taylor and McKeon, 1997*).

So far, only few substrates of Bub1 have been identified. Best characterized is the phosphorylation of histone H2A on threonine 120 (T120) (*Kawashima et al., 2010*). Phosphorylation of this site by Bub1 can be demonstrated not only in vitro but also in living cells (*Kawashima et al., 2010*; *Lin et al., 2014*; *Sharp-Baker and Chen, 2001*). Histone H2A phosphorylation on T120 triggers the centromere localization of Sgo1, which in turn recruits the CPC subunit Borealin (*Kawashima et al., 2010*; *Liu et al., 2013*; *Tsukahara et al., 2010*; *Yamagishi et al., 2010*). Centromere recruitment of the CPC is further enhanced by the kinase Haspin, which phosphorylates histone H3 at T3 and triggers the centromere binding of the CPC component Survivin (*Du et al., 2012*; *Kelly et al., 2010*; *Wang et al., 2010*). Another intriguing potential substrate of Bub1 is the APC/C co-activator Cdc20 (*Lin et al., 2014*; *Tang et al., 2004a*). Whether Bub1 phosphorylates Cdc20 in living cells remains to be determined, but recent studies strongly suggest that Bub1 binding to Cdc20 is important for SAC function (*Di Fiore et al., 2015*; *Vleugel et al., 2015*).

Genetic, biochemical or siRNA-mediated depletion of Bub1 protein clearly interferes with chromosome alignment and SAC activity, but the importance of Bub1 catalytic activity has long been subject to debate (*Bolanos-Garcia and Blundell, 2011*; *Elowe, 2011*; *Funabiki and Wynne, 2013*). For example, while a Bub1 mutant completely lacking the kinase domain is checkpoint proficient in *Saccharomyces cerevisiae* (*Fernius and Hardwick, 2007*), conflicting data have been reported on the importance of Bub1 kinase activity in fission yeast *Schizosaccharomyces pombe* (*Rischitor et al., 2007*; *Vanoosthuyse et al., 2004*; *Yamaguchi et al., 2003*). Similarly, in *Xenopus* egg extracts, catalytically inactive Bub1 can sustain the SAC (*Sharp-Baker and Chen, 2001*), although kinase-proficient Bub1 may be more efficient (*Boyarchuk et al., 2007*; *Chen, 2004*). In mammalian cells, several studies point to the conclusion that Bub1 mutants devoid of catalytic activity are able to restore many, albeit not all, aspects of chromosome congression and SAC function (*Klebig et al., 2009*; *McGuinness et al., 2009*; *Perera and Taylor, 2010a*; *Ricke et al., 2012*).

To address the role of Bub1 kinase activity in mammalian mitosis, we have made use of two novel small molecule inhibitors, BAY-320 and BAY-524. Using biochemical and cellular assays, we show that these ATP-competitive inhibitors potently and specifically block human Bub1 both in vitro and

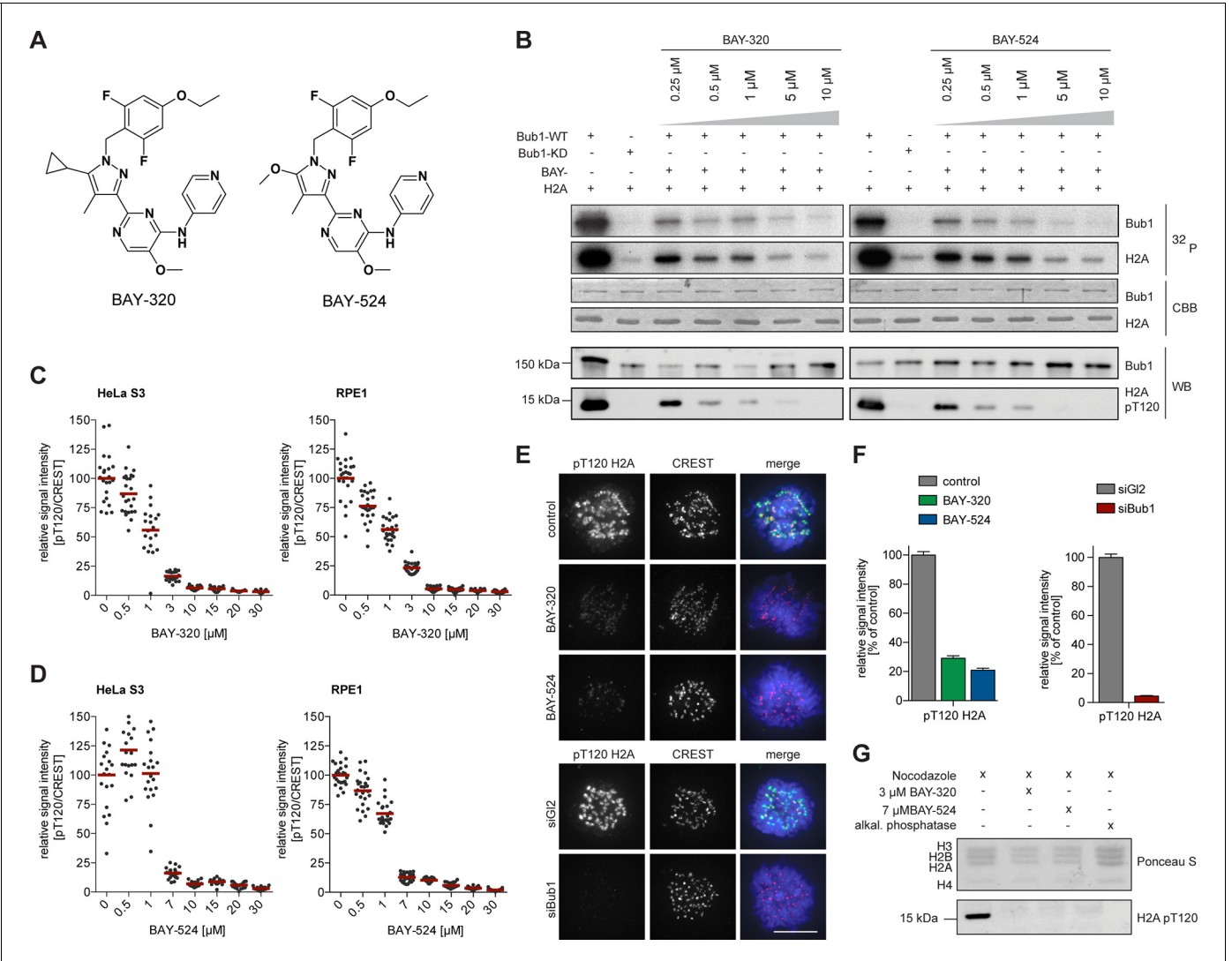

**Figure 1.** BAY-320 and BAY-524 inhibit Bub1 kinase. (A) Chemical structure of ATP-competitive inhibitors BAY-320 and BAY-524. (B) In vitro kinase assays showing dose-dependent inhibition of Bub1 kinase activity towards histone H2A. The assays were performed by mixing human wild-type (WT) or kinase-dead (KD) LAP-Bub1, ectopically expressed in and purified from mitotic HEK 293T cells, with recombinantly expressed histone H2A as a substrate, $\gamma$-$^{32}$P-ATP and increasing doses of the Bub1 inhibitors BAY-320 and BAY-524. After 30 min at 30°C, reactions were stopped and analyzed by gel electrophoresis. Bub1 autophosphorylation and H2A phosphorylation were visualized by autoradiography ($^{32}$P) and protein levels monitored by Coomassie brilliant blue staining (CBB). Histone H2A-T120 phosphorylation (pT120-H2A) was detected by phospho-antibody probing of Western blots (WB) and Bub1 was monitored as control. (C, D) Inhibition of Bub1 reduces histone H2A-T120 phosphorylation. Asynchronous cultures of HeLa S3 (left panels) and RPE1 cells (right panels) were treated with the proteasomal inhibitor MG132 for 2 hr, followed by the addition of 3.3 µM nocodazole and increasing doses of BAY-320 (C) or BAY-524 (D) for 1 hr. The cells were fixed and analyzed by immunofluorescence microscopy (IFM). Scatter plots show centromeric levels of pT120-H2A (n = 19–28 cells per condition). Bars represent mean values. (E) HeLa S3 cells were synchronized by thymidine block, released for 10 hr in the presence of solvent (control), 3 µM BAY-320 or 7 µM BAY-524 and analyzed by quantitative IF (top panels). Cells transfected with mock (Gl2) or Bub1 siRNA-oligonucleotides for 48 hr were synchronized and analyzed in parallel (bottom panels). The cells were stained with antibodies raised against Bub1 and pT120-H2A. Human CREST serum was used to identify centromeres and DNA was stained with DAPI; scale bars represent 10 µm. (F) Histograms showing the average signal intensities of centromeric pT120-H2A observed in the experiments described in (E); n = 73–107 cells per condition. Error bars represent standard error of the mean (SEM). (G) To monitor the efficacy of Bub1 kinase inhibition within cells, HeLa S3 cells were synchronized by thymidine block and released for 14 hr in the presence of 3.3 µM nocodazole as well as Bub1 inhibitors or solvent as indicated. Prometaphase-arrested cells were harvested by shake-off and mitotic cell extracts were treated with or without phosphatase inhibitor for 30 min at 30°C. Histone isolation was followed by Western blot analysis of pT120-H2A. Equal loading was monitored by Ponceau S staining.

The following figure supplement is available for figure 1:

**Figure supplement 1.** BAY-320 and BAY-524 inhibit Bub1 kinase.

in living cells. By comparing phenotypes provoked by Bub1 kinase inhibition and Bub1 protein depletion, we are able to differentiate between catalytic and non-catalytic functions of Bub1. Our data indicate that Bub1 catalytic activity is largely dispensable for chromosome alignment and SAC function, arguing that Bub1 largely operates as a scaffolding protein. However, even though Bub1 inhibition per se exerts only minor effects on mitotic fidelity, BAY-320 and BAY-524 treatment sensitizes cells to clinically relevant low doses of Paclitaxel, resulting in remarkable impairment of chromosome segregation and cell proliferation.

## Results

### BAY-320 and BAY-524 specifically inhibit Bub1 kinase

The chemical synthesis of small molecule inhibitors against Bub1 has recently been described (*Hitchcock et al., 2013*). In this study, we used the two substituted benzylpyrazole compounds, 2-[5-cyclopropyl-1-(4-ethoxy-2,6-difluorobenzyl)-4-methyl-1H-pyrazol-3-yl]-5-methoxy-N-(pyridin-4-yl)pyrimidin-4-amine and 2-[1-(4-ethoxy-2,6-difluorobenzyl)-5-methoxy-4-methyl-1H-pyrazol-3-yl]-5-methoxy-N-(pyridin-4-yl)pyrimidin-4-amine, abbreviated as BAY-320 and BAY-524, respectively (*Figure 1A*). In vitro inhibition of Bub1 by BAY-320 and BAY-524 was demonstrated by monitoring both Bub1 autophosphorylation and phosphorylation of histone H2A on T120 (*Kawashima et al., 2010*) (*Figure 1B*). In presence of 2 mM ATP, both compounds inhibited the recombinant catalytic domain of human Bub1 (amino acids 704–1085) with an $IC_{50}$ of 680 ± 280 nM and 450 ± 60 nM, respectively (*Supplementary file 1*). When tested against a panel of 222 protein kinases, BAY-320 showed only modest cross reactivity with other kinases, even when used at a concentration of 10 μM (*Supplementary file 2*). Furthermore, quantitative measurements of BAY-320 interactions with 403 human kinases, using an active site-directed competition-binding assay, showed exquisite binding selectivity for Bub1 (*Supplementary file 3*).

To test whether BAY-320 and BAY-524 also inhibit Bub1 in intact cells, increasing doses of inhibitors were applied to mitotically synchronized hTERT-RPE1 (RPE1) and HeLa cells, and phospho-histone H2A-T120 staining at kinetochores was monitored by immunofluorescence (*Figure 1C–F* and *Figure 1—figure supplement 1A, B*) and in-cell western assays (*Figure 1—figure supplement 1C*). These studies revealed that near-maximal inhibition of Bub1 could be achieved by using BAY-320 at 3–10 μM and BAY-524 at 7–10 μM and these concentrations were therefore used in all future experiments on intact cells. To corroborate the above immunofluorescence data, histones were purified from control and inhibitor-treated cells. Examination of histone H2A phosphorylation by Western blotting revealed that treatment of cells with either BAY-320 or BAY-524 drastically reduced T120 phosphorylation (*Figure 1G*). Thus, BAY-320 and BAY-524 act as potent and selective inhibitors of Bub1 kinase in both biochemical and cellular assays and thus constitute attractive tools to study Bub1 catalytic function during mitosis.

### Impact of Bub1 kinase inhibition and Bub1 depletion on mitotic progression

Next, we set out to directly compare the impact of Bub1 kinase inhibition with the previously reported consequences of Bub1 depletion (*Boyarchuk et al., 2007*; *Johnson et al., 2004*; *Kitajima et al., 2005*; *Klebig et al., 2009*; *Logarinho et al., 2008*; *Meraldi and Sorger, 2005*; *Tang et al., 2004b*) or genetic Bub1 knock-out (*Jeganathan et al., 2007*; *Perera and Taylor, 2010a*; *Perera et al., 2007*; *Ricke et al., 2012*). In a first series of experiments, we used time-lapse imaging to compare progression through mitosis in asynchronously growing HeLa and RPE1 cells in response to either Bub1 inhibition or siRNA-mediated Bub1 depletion. In line with previous results (*Kitajima et al., 2005*; *Tang et al., 2004b*), depletion of Bub1 from HeLa cells significantly prolonged duration of mitosis, due to delayed chromosome alignment and delays in prometa- and metaphase (*Figure 2A and C*, *Figure 2—figure supplement 1A*). In stark contrast, treatment with either BAY-320 or BAY-524 provoked at most minor effects on mitotic progression, marked by a short delay of anaphase onset (*Figure 2B and C*, *Figure 2—figure supplement 1B and C*). Furthermore, in contrast to aneuploid HeLa cells, diploid RPE1 cells were not significantly affected by either Bub1 inhibition or depletion (*Figure 2D* and *Figure 2—figure supplement 1D*). Efficiency of siRNA-mediated depletion was monitored by Western blotting (*Figure 2—figure supplement 1E*). Flow-

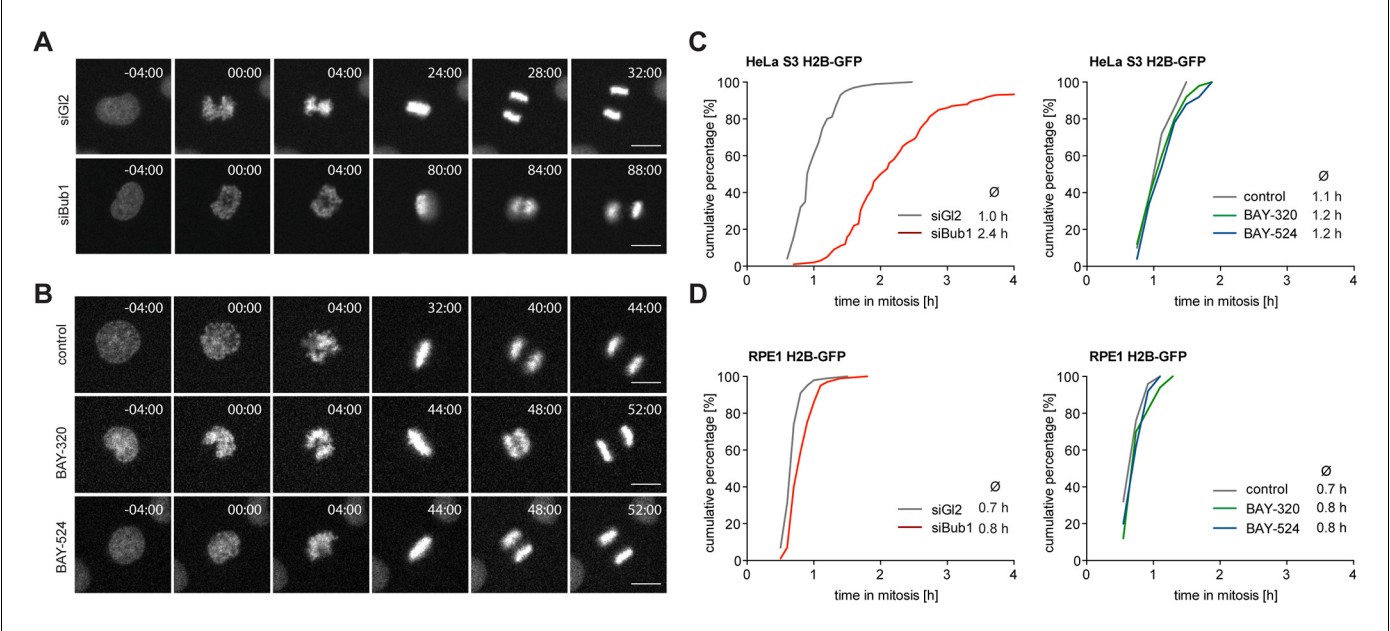

**Figure 2.** Inhibition of Bub1 kinase activity barely affects mitotic progression. (**A, B**) Representative stills from time-lapse recordings of asynchronously growing cultures of HeLa S3 cells stably expressing GFP-tagged histone H2B. The cells were either treated with Bub1 inhibitors (3 μM BAY-320 and 7 μM BAY-524) or transfected with control (Gl2) or Bub1 siRNA for 48 hr prior to time-lapse microscopy. Scale bars represent 10 μm. (**C, D**) Graphs show the cumulative frequency of mitotic duration determined by cell rounding/flattening. Indicated averages represent the time spent in mitosis (n = 100 cells per condition).

The following figure supplement is available for figure 2:

**Figure supplement 1.** Inhibition of Bub1 kinase activity barely affects mitotic progression.

cytometric analyses confirmed that Bub1 depletion from HeLa cells causes an increase in the G2/M population of HeLa but not RPE1 cells and that Bub1 inhibition by BAY-320 or BAY-524 did not detectably affect cell cycle profiles in either cell line (*Figure 2—figure supplement 1F and G*). We conclude that the inhibition of Bub1 kinase activity in either HeLa or RPE1 cells produces at most subtle effects on mitotic progression, whereas Bub1 depletion exerts more profound effects, at least in HeLa cells. These results are consistent with the demonstration that Bub1 kinase activity is not required for the development and viability of mice (*Perera and Taylor, 2010b*; *Ricke et al., 2012*).

## Bub1 kinase regulates Shugoshin localization and chromatid cohesion

One of the most interesting effects of Bub1 depletion described so far relates to sister chromatid cohesion (*Boyarchuk et al., 2007*; *Fernius and Hardwick, 2007*; *Kitajima et al., 2005*; *Tang et al., 2004b*). In particular, depletion of Bub1 was shown to cause persistent arm cohesion and a redistribution of Sgo proteins from centromeres to chromosome arms (*Kitajima et al., 2005*). To directly demonstrate a role for Bub1 kinase activity in sister chromatid cohesion, we analyzed chromosome spreads prepared from mitotic HeLa cells or RPE1 cells after treatment with Bub1 inhibitors (*Figure 3A*, *Figure 3—figure supplement 1A*) or Bub1-specific siRNA for comparison (*Figure 3B* and *Figure 3—figure supplement 1B*). While mitotic chromosome spreads from nocodazole-treated control cells showed the expected X-shape structure, indicative of centromere cohesion, most cells treated with either Bub1 inhibitors or Bub1 siRNA showed sister chromatids whose arms remained paired (*Figure 3A–C* and *Figure 3—figure supplement 1*). Moreover, centromeric levels of Sgo1 and Sgo2 were reduced to ~20% of control values in BAY-320 or BAY-524 treated cells (*Figure 3D and E*) and, concomitantly, a significant redistribution of Sgo2 to chromosome arms could be observed (*Figure 3F and G*). We thus conclude that Bub1 catalytic activity contributes to the regulation of sister chromatid cohesion and the localization of Sgo proteins.

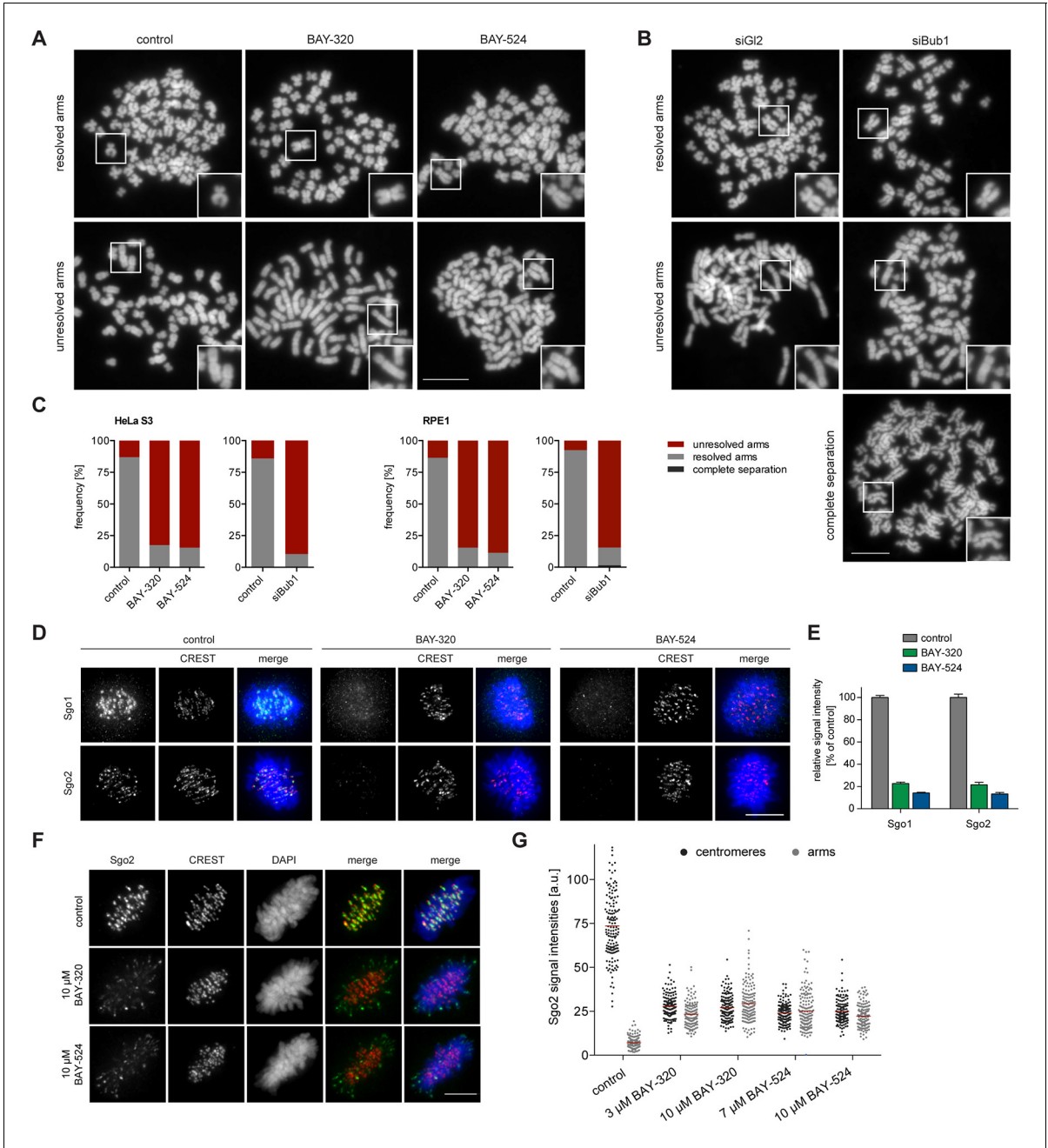

**Figure 3.** Inhibition of Bub1 affects Sgo1 and Sgo2 localization and chromatid cohesion. (**A, B**) HeLa S3 cells were synchronized by thymidine block and released for 12 hr in the presence of 3.3 μM nocodazole as well as solvent (control), 3 μM BAY-320 or 7 μM BAY-524. Cells transfected with mock (Gl2) or Bub1 siRNA-oligonucleotides for 48 hr were synchronized and analyzed in parallel. Micrographs show representative chromosome spreads prepared from mitotic cells. Insets show magnifications of chromosomes; they illustrate representative chromatid cohesion states. (**C**) Quantification of results of the experiments described in (**A**) and (**B**) as well as *Figure 3—figure supplement 1*; n = 200 cells per condition. (**D**) HeLa S3 cells were released from a thymidine arrest into solvent, 3 μM BAY-320 or 7 μM BAY-524. The cells were fixed and stained for Sgo1, Sgo2, CREST and DNA (DAPI) and analyzed by IFM. Scale bars represent 10 μm. (**E**) Histogram showing average centromeric Sgo levels observed in the experiments described in (**A**); n = 43–120 cells per condition. Error bars represent SEM. (**F**) Asynchronous cultures of RPE1 cells were treated with indicated doses of Bub1 inhibitors for 3 hr, fixed and analyzed by IFM. Scale bar represents 5 μm. (**G**) Dot plot showing the quantitative results of the experiment shown in (**F**). Sgo2 levels at centromeres and chromosome arms were determined in metaphase cells (n = 150 centromere/arm regions from 15 different cells). Bars represent mean values.

The following figure supplement is available for figure 3:

**Figure supplement 1.** Inhibition of Bub1 affects chromatid cohesion.

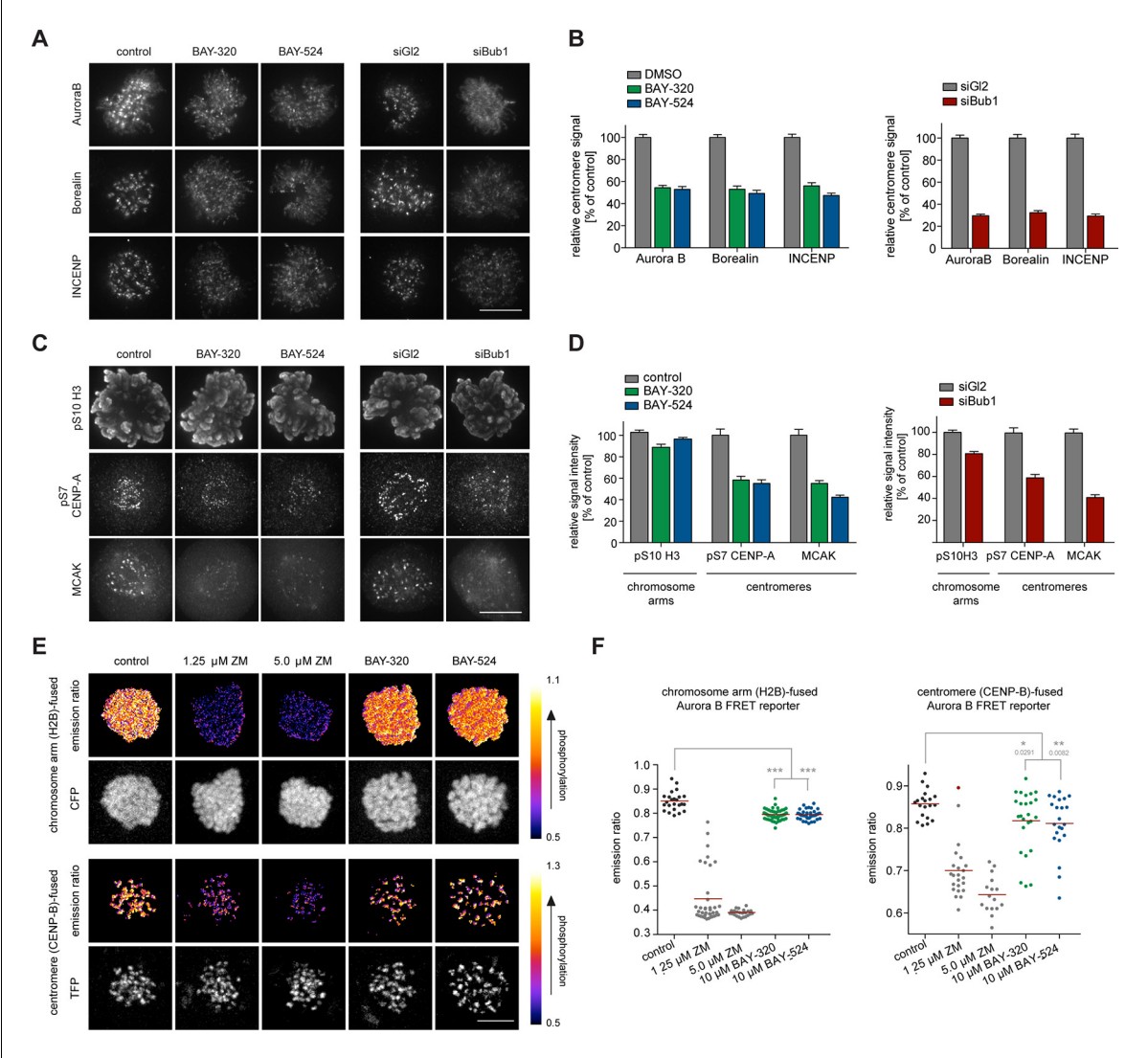

**Figure 4.** Bub1 inhibition affects localization and activity of the CPC. (A, C) Untreated or siRNA transfected (siBub1, siGl2 for control) HeLa S3 cells were synchronized by thymidine block and released for 10 hr, as indicated (BAY-320 was used at 3 μM, BAY-524 at 7 μM). Cells were fixed and stained for Aurora B, Borealin, INCENP, pS7-CENP-A, pS10-histone H3, MCAK, CREST and DNA (DAPI) and analyzed by IFM. Scale bars represent 10 μm. (B, D) Histograms show quantitative results of the experiments described in (A, C). Measurements represent centromeric levels except for pS10-histone H3 signals, which was monitored along chromosome arms (n = 40–113 cells per condition). Scale bars represent 10 μm, error bars represent SEM. (E) FRET experiments were performed on HeLa Kyoto cells stably expressing chromatin (H2B)- or centromere (CENP-B)-fused FRET reporters for Aurora B activity. Cells were synchronized in mitosis by 6 hr treatment with 3.3 μM nocodazole, before the indicated inhibitors and 20 μM MG132 were added prior to live fluorescence microscopy. Heat-map represents the phosphorylation status of the reporter. Scale bar represents 10 μm. (F) Left panel: scatter plot depicts CFP/FRET emission ratios of reporter targeted to chromatin (H2B; n = 23–52 cells per condition). Right panel: scatter plot depicts TFP/FRET emission ratios of reporter targeted to centromeres (CENP-B, n = 16–34 cells per condition). Bars represent mean values; ***p<0.001 (from unpaired two-tailed Student's t-test).

The following figure supplement is available for figure 4:

**Figure supplement 1.** Bub1 inhibition affects localization and activity of the CPC.

## Bub1 inhibition affects CPC localization

In addition to preserving sister chromatid cohesion, Sgo1 and Sgo2 play important roles in the recruitment of the CPC, comprising Aurora B kinase (*Kawashima et al., 2007*; *Tsukahara et al., 2010*). This prompted us to investigate the impact of Bub1 inhibition on Aurora B localization and

activity. Consistent with the marked effects on centromere localization of Sgo1/2, we also observed significant effects of Bub1 inhibition on Aurora B localization. After treatment of HeLa cells with BAY-320 or BAY-524, all CPC subunits examined were partially displaced from centromeres (*Figure 4A and B*, *Figure 4—figure supplement 1A*). While Bub1 inhibition reduced centromeric levels of Aurora B, Borealin and INCENP by ~50% (*Figure 4A and B*, left panels), depletion of Bub1 lowered centromere levels of these CPC components by ~70% (*Figure 4A and B*, right panels). We emphasize that, due to a lack of resolution, these experiments do not discriminate between centromere- and KT-associated pools of the CPC.

To examine the impact of Bub1 inhibition on the catalytic activity of Aurora B at both centromeres and chromosome arms, we next monitored phosphorylation of CENP-A Ser7 (*Zeitlin et al., 2001*) and histone H3 Ser10 (*Hirota et al., 2005*; *Hsu et al., 2000*), respectively. Compared to control cells, both Bub1 inhibition and depletion reduced CENP-A and histone H3 phosphorylation by ~50% and ~10–20%, respectively (*Figure 4C and D*, *Figure 4—figure supplement 1B*), suggesting that interference with Bub1 primarily affects Aurora B activity at centromeres. This conclusion was corroborated by showing that both inhibition and depletion of Bub1 reduced the centromere association of the Aurora B effector protein MCAK (*Andrews et al., 2004*) by ~50% (*Figure 4C and D*). Furthermore, use of biosensors for Aurora B activity (*Fuller et al., 2008*) revealed a reduction in fluorescence resonance energy transfer (FRET) ratios for sensors tethered to either chromosome arms (through fusion to H2B) or centromeres (through fusion to histone CENP-B) (*Figure 4E and F*, *Figure 4—figure supplement 1C*). Collectively, these observations demonstrate that Bub1-dependent phosphorylation plays a major role in the regulation of Aurora B localization and activity. However, neither Bub1 inhibition nor Bub1 depletion resulted in complete removal of Aurora B from centromeres, prompting us to examine the relative contributions of Bub1 and Haspin to the process of CPC recruitment.

## Bub1 and Haspin cooperate to recruit CPC to centromeres

While inhibition of Bub1 by BAY-320 or BAY-524 or inhibition of Haspin by 5-Iodotubercidin (*De Antoni et al., 2012*) similarly reduced centromere levels of the CPC components Aurora B, Borealin and INCENP to ~40%, combined inhibition of both kinases resulted in a ~80% reduction in CPC levels at centromeres (*Figure 5A and B*, *Figure 5—figure supplement 1A*). As an important control, treatment of cells with only BAY-320 or BAY-524 did not detectably affect the phosphorylation of the Haspin substrate histone H3 (T3), attesting to the specificity of the two Bub1 inhibitors (*Figure 5A and B*).

To quantify CPC localization over chromosome arms, analysis of fixed cells proved inadequate. We therefore used an RPE1 cell line expressing one endogenous allele of Aurora B tagged with EGFP (*von Schubert et al., 2015*) to monitor the subcellular localization of this kinase in living cells. Following Bub1 inhibition, Aurora B-EGFP levels at chromosome arms increased approximately twofold, concomitant with the described reduction of Aurora B at centromeres (*Figure 5C and D*) (*Boyarchuk et al., 2007*; *Ricke et al., 2012*). Interestingly, this change in localization showed a strong correlation with the redistribution of Sgo2 (*Figure 5—figure supplement 1B and C*). In contrast, treatment of cells with the Haspin inhibitor 5-Iodotubercidin did not induce any significant redistribution of Aurora B from centromeres to chromosome arms; instead, inhibition of Haspin caused an overall reduction of EGFP signals at both centromeres and chromosome arms (*Figure 5C and D*). Combined inhibition of Bub1 and Haspin displaced Aurora B from both centromeres and chromosome arms (*Figure 5C and D*), in line with the analysis of fixed cells described above. Taken together, these data corroborate the notion that Bub1 and Haspin cooperate in the recruitment of CPC to centromeres through phosphorylation of histone H2A-T120 and histone H3-T3, respectively. In addition, they reveal a role for Bub1 kinase activity, but not Haspin, in restricting CPC localization to the centromere.

Considering the role of Aurora B kinase in the regulation of KT-MT interactions and SAC signaling, the above results raised the question of what contributions Bub1 activity might possibly make to chromosome congression and/or the SAC. Although our initial analyses had not revealed a major impact of BAY-320 or BAY-524 on the overall timing of mitotic progression (*Figure 2*), we considered the possibility that inhibition of Bub1 might provoke compensatory effects on mitotic timing, notably a delay in congression and a concomitant acceleration of mitotic exit. According to such a scenario, effects on timing might conceivably cancel each other. In support of this possibility, we

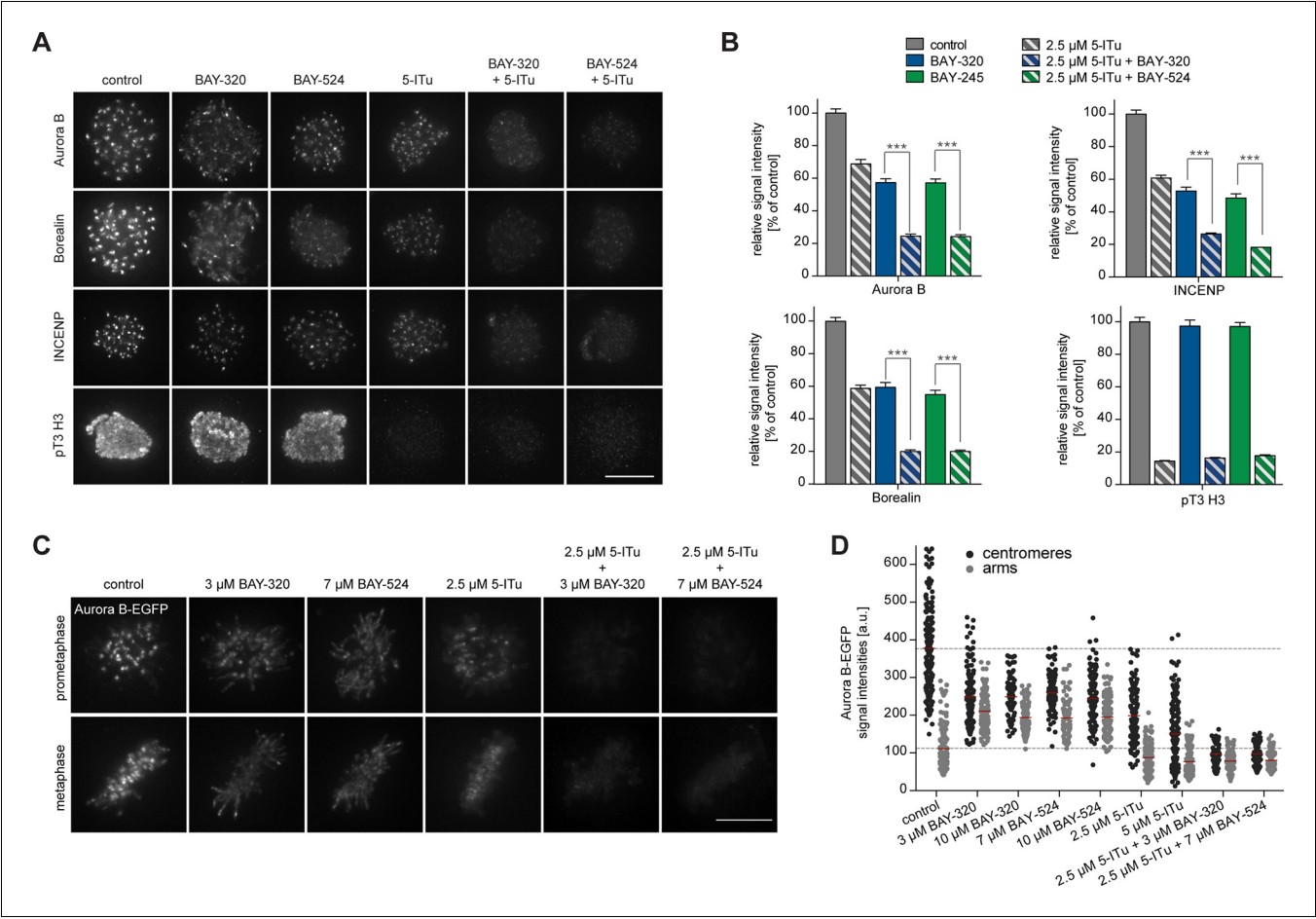

**Figure 5.** Bub1 and Haspin inhibition exert additive effect on centromere association of CPC. (**A**) HeLa S3 cells were released from a thymidine block into 3.3 μM nocodazole, before they were additionally treated for 2 hr with the proteasomal inhibitor MG132 and indicated kinase inhibitors. The Haspin inhibitor 5-iodotubercidin (5-ITu [*De Antoni et al., 2012*]) was used at a concentration of 2.5 μM, BAY-320 at 3 μM and BAY-524 at 7 μM. Cells were fixed, stained for pT3-H3, Aurora B, Borealin, INCENP, CREST and DNA (DAPI) and analyzed by IFM. Anti-pT3-H3 antibody was used to monitor Haspin and Bub1 inhibition, respectively. Scale bar represents 10 μm. (**B**) Histograms show average centromeric (AurB, Borealin, INCENP) or chromosome arm (pT3-H3) signal intensities observed in the experiments shown in (**A**); n = 20–100 cells per condition. Error bars represent SEM, ***p < 0.001 (from unpaired two-tailed Student's t-test). (**C**) RPE1 cells expressing endogenously EGFP-tagged Aurora B were incubated with the indicated drugs for several hours before EGFP signals were recorded by live fluorescence imaging. Scale bar represents 5 μm. (**D**) Scatter plots depict Aurora B-EGFP signal intensities at centromeres or arms after treatment with indicated drugs (n = 84–185 centromeres/arm regions from 5–6 cells per condition). Bars represent mean values. For comparison, dashed lines mark the mean values of Aurora B-EGFP signal intensities at arms and centromeres in control cells. Measurements relate to the experiment shown in (**C**).

The following figure supplement is available for figure 5:

**Figure supplement 1.** Bub1 and Haspin inhibition exert additive effect on centromere association of CPC.

emphasize that the inhibition of mitotic kinases with pleiotropic functions have previously been shown to provoke opposing phenotypes (*Santaguida et al., 2011*; *von Schubert et al., 2015*). To explore the possibility of compensatory effects of Bub1 inhibition, we thus carried out more detailed analyses of mitotic progression, notably KT-recruitment of SAC components, SAC signaling and chromosome congression.

## Bub1 inhibition produces minor effects on SAC signaling in HeLa or RPE1 cells

Depletion of Bub1 is known to weaken SAC signaling in human cells (*Klebig et al., 2009*; *Meraldi and Sorger, 2005*; *Perera et al., 2007*). To test the impact of Bub1 catalytic activity on

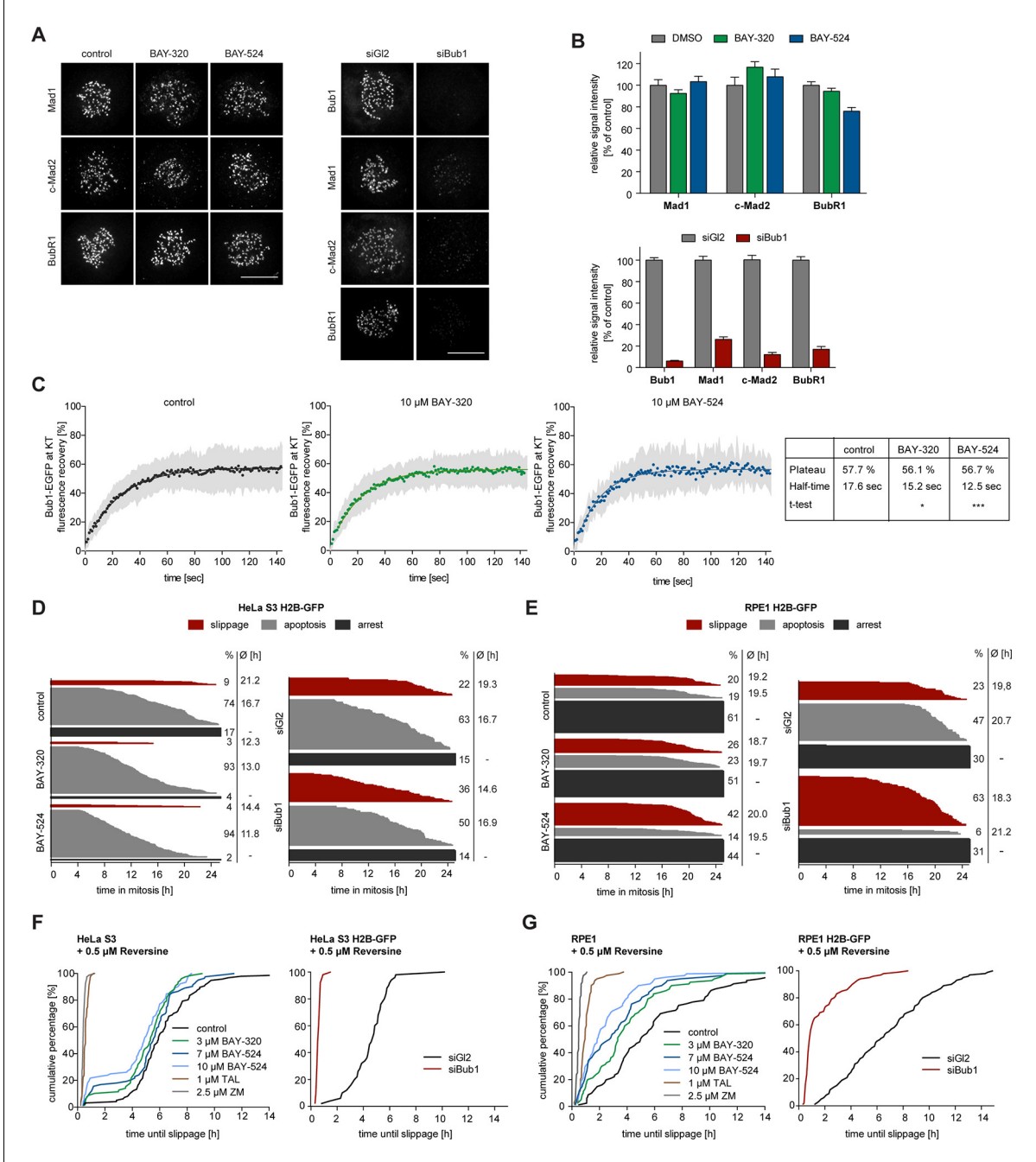

**Figure 6.** Bub1 inhibition marginally affects SAC signaling. (**A**) Inhibition of Bub1 kinase does not significantly affect recruitment of SAC effectors to unattached KTs. HeLa S3 cells were synchronized by thymidine block and released for 10 hr in the presence of solvent (control), 3 µM BAY-320 or 7 µM BAY-524. Cells transfected with mock (Gl2) or Bub1 siRNA-oligonucleotides for 48 hr were synchronized and analyzed in parallel. The cells were fixed and stained for Bub1, Mad1, closed Mad2 (C-Mad2), CREST and DNA (DAPI) and analyzed by IFM. (**B**) Histogram shows average KT levels of indicated proteins (n = 20–50 cells per condition) observed in the experiment shown in (**A**). Error bars represent SEM. (**C**) RPE1 cells expressing endogenously tagged Bub1-EGFP were synchronized in mitosis by overnight treatment with the Eg5 inhibitor STLC (10 µM) and subsequently treated with 3.3 µM nocodazole and 20 µM MG132 as well as solvent (control), 10 µM BAY-320 or 10 µM BAY-524. Bub1-EGFP KT levels were recorded by 1 sec time-lapse microscopy. After 5 sec, a single KT pair was bleached and fluorescence recovery was monitored. Traces illustrate average fluorescence recovery at KT pairs (n = 10–16 KT pairs per condition); shaded areas represent standard deviation (SD). Half-times and plateaus were determined by non-linear curve fitting based on a one-phase association. (**D, E**) Asynchronously growing cultures of HeLa S3 (**D**) or RPE1 (**E**) cells stably expressing GFP-tagged histone H2B were either directly treated with 3.3 µM nocodazole and the kinase inhibitors BAY-320 (3 µM) and BAY-524 (7 µM) or transfected with control (Gl2) or Bub1 siRNA for 48 hr prior to addition of nocodazole. Cell fates (continued arrest, apoptosis or slippage) and duration of mitotic arrest were determined by fluorescence time-lapse imaging (n = 150 cells per condition, accumulated from 3 independent experiments). Frequencies of observed

*Figure 6 continued on next page*

*Figure 6 continued*

cell fates as well as average times of arrest are indicated. (**F**) Asynchronously growing HeLa S3 cells or HeLa cells stably expressing GFP-tagged histone H2B were treated with 3.3 µM nocodazole and 0.5 µM of the Mps1 inhibitor Reversine as well as solvent (control), 3 and 10 µM BAY-320, 7 and 10 µM BAY-524 or 2.5 µM of the Aurora B inhibitor ZM-447439 (ZM) (left panel). Alternatively, cells were transfected with control (Gl2) or Bub1 siRNA oligonucleotides for 48 hr prior to addition of 3.3 µM nocodazole and 0.5 µM Reversine (right panel). The cells were monitored by fluorescence time-lapse microscopy and the time elapsed from nuclear envelope breakdown to SAC override and mitotic slippage was determined. Traces illustrate the cumulative frequency of mitotic duration before slippage (n = 50 cells per condition). (**G**) Asynchronously growing RPE1 cells stably expressing GFP-tagged histone H2B were treated and analyzed as described in (**F**). Scale bars represent 10 µm.

The following figure supplements are available for figure 6:

**Figure supplement 1.** Panels relate to the quantitative data shown in *Figure 6D*.

**Figure supplement 2.** Tagging strategy and IFM analysis of Bub1 KT levels.

SAC function, we first analyzed KT levels of Mad1, Mad2 and BubR1 in BAY-320 or BAY-524 treated cells. With the possible exception of a very minor effect on BubR1, the localization of none of these SAC proteins was significantly affected by Bub1 inhibition (*Figure 6A and B*, *Figure 6—figure supplement 1A*). In sharp contrast, and in agreement with previous reports (*Boyarchuk et al., 2007*; *Johnson et al., 2004*; *Overlack et al., 2015*; *Sharp-Baker and Chen, 2001*), Bub1 depletion decreased KT recruitment of all three proteins by 80–90% (*Figure 6A and B*, *Figure 6—figure supplement 1B*). Thus, the recruitment of several SAC components to KTs strongly depends on Bub1 protein, but not Bub1 kinase activity.

The association of Bub1 with unattached KTs is dynamic (*Howell et al., 2004*), raising the question of how Bub1 turnover at KTs is regulated. In the case of the SAC kinase Mps1, autophosphorylation constitutes a major mechanism for controlling Mps1 levels at KTs (*Hewitt et al., 2010*; *Jelluma et al., 2010*; *von Schubert et al., 2015*), and a recent study suggests that Bub1 turnover at KTs is also regulated by autophosphorylation (*Asghar et al., 2015*). To determine whether Bub1 dynamics at KTs is affected by inhibition of Bub1 activity, we made use of an RPE1 cell line harboring one allele of Bub1 tagged by EGFP at the endogenous locus (*Figure 6C*, *Figure 6—figure supplement 2A*). After the treatment of cells with nocodazole to assure complete MT depolymerization and full SAC activation (*Santaguida et al., 2011*; *Yang et al., 2009*), Bub1 levels and turnover at KTs were measured by immunofluorescence microscopy and fluorescence recovery after photobleaching (FRAP), respectively. In comparison to control cells, neither BAY-320 nor BAY-524 detectably affected steady-state Bub1 levels at KTs (*Figure 6—figure supplement 2B and C*), in line with a recent report (*Liu et al., 2015*). More importantly, FRAP experiments revealed only minor effects of Bub1 inhibition on Bub1 dynamics at KTs (*Figure 6C*). The extent of fluorescence recovery after FRAP was not significantly different in control cells and inhibitor treated cells, revealing an immobile fraction of ~42%, in excellent agreement with previous data (*Asghar et al., 2015*; *Howell et al., 2004*). The half-time of Bub1 recovery at KTs after FRAP was ~18 sec in controls, again in good agreement with previous data (*Asghar et al., 2015*; *Howell et al., 2004*). However, whereas Asghar and colleagues observed a ~50% reduction in the half-time of recovery of an exogenously expressed, catalytically inactive EGFP-Bub1 mutant, we found that recovery of endogenously tagged wild-type EGFP-Bub1 was only marginally accelerated by Bub1 inhibition (half-time reduced from 18 s to 12–15 s) (*Figure 6C*). When considering this discrepancy, it is important to bear in mind that our data reflect turnover of chemically inhibited wild-type Bub1 expressed at endogenous levels, whereas Ashghar and colleagues monitored mutant versions of overexpressed Bub1, raising the possibility that their results may have been influenced by expression levels and/or mutation-induced structural alterations. We conclude that the effects of Bub1 activity on Bub1 turnover at KTs are at most minor, particularly when compared to the striking effects of Mps1 activity on Mps1 dynamics at KTs (*Hewitt et al., 2010*; *Jelluma et al., 2010*; *von Schubert et al., 2015*).

As a further read-out for the effects of Bub1 inhibition on SAC activity, we used live cell imaging to monitor the responses of nocodazole-arrested HeLa and RPE1 cells to BAY-320 or BAY-524 and compared these to the responses seen in Bub1-depleted cells (*Figure 6D and E*). Over a 24 hr observation period, the percentage of HeLa cells maintaining a SAC arrest dropped from 17% in

controls to 4% and 2% in response to Bub1 inhibition by BAY-320 and BAY-524, respectively (*Figure 6D*, left panel). These shifts in cell fates were largely compensated by increases in the percentages of cells undergoing apoptosis, from 74% in controls to 94% in Bub1-inhibited cells. In contrast, although the duration of mitosis was slightly reduced upon Bub1 inhibition, the extent of mitotic slippage remained at less than 10% under all conditions. In RPE1 cells, maintenance of SAC arrest over 24 hr was more pronounced, but again the percentage of arrested cells dropped from 61% in controls to 51%/44% in response to Bub1-inhibition, with increasing proportions of cells undergoing apoptosis or mitotic slippage (*Figure 6E*, left panel). For comparison, depletion of Bub1 from either HeLa or RPE1 cells resulted in a 2–3 fold increase in mitotic slippage at the expense of apoptosis, while the proportion of cells sustaining an arrest remained roughly constant (*Figure 6D and E*, right panels). Collectively, these results indicate that Bub1 activity contributes to the maintenance of maximal SAC activity, but that Bub1 protein levels are more important, most likely reflecting the observed role of Bub1 in the KT recruitment of SAC components (*Figure 6A*).

Importantly, we also compared the requirements for Bub1 activity and Bub1 protein in a cellular background in which SAC activity was partially compromised by the treatment of HeLa or RPE1 cells with a low dose of Reversine, a widely used inhibitor of the SAC kinase Mps1 (*Santaguida et al., 2010*). In agreement with the results described above, Bub1 inhibition marginally reduced the time that Reversine-treated cells remained arrested before overriding nocodazole-induced arrest (*Figure 6F and G*, left panels). Addition of Aurora B or Plk1 inhibitors, used as positive controls, led to the expected shortening of the duration of mitotic arrest (*Figure 6F and G*, left panels) (*Saurin et al., 2011*; *von Schubert et al., 2015*). Similarly, Bub1 depletion also caused a drastic shortening of arrest (*Figure 6F and G*, right panels). Taken together with previous studies (*Klebig et al., 2009*; *Perera and Taylor, 2010a*; *Perera et al., 2007*; *Ricke et al., 2012*), these observations demonstrate that the scaffolding function of Bub1 is required for the SAC, but its catalytic activity is largely dispensable.

## Bub1 inhibition does not significantly impair chromosome congression

To analyze the impact of Bub1 inhibition on chromosome alignment, we treated cells with the Eg5 inhibitor Monastrol (*Kapoor et al., 2000*) and then monitored the restoration of KT-MT attachments during spindle bipolarization in response to drug washout (*Figure 7A and B*). While nearly 28% of Bub1-depleted cells failed to completely align all chromosomes, more than 90% of Bub1-inhibited cells showed complete alignment that was indistinguishable from control cells. Inhibition of Aurora B, analyzed for control, resulted in the expected impairment of alignment (*Figure 7A and B*). To complement these assays, we also used immunofluorescence microscopy to quantify the frequency of micronucleation, a read-out for chromosome segregation errors, in HeLa and RPE1 cells. While partial inhibition of Aurora B kinase provoked an increase in micronucleation in both cell lines, as expected (*Gohard et al., 2014*; *Tao et al., 2009*), Bub1 inhibition only marginally increased the frequency of micronucleation (*Figure 7C*). This result supports the view that Bub1 inhibition causes surprisingly mild defects in chromosome congression or segregation (*Figure 2* and *7A*). Further corroborating this conclusion, we found that BAY-320 or BAY-524 treatment exerted no significant effects on the kinetochore recruitment of the motor protein CENP-E (*Figure 7D and E*, left panels). In contrast, Bub1 depletion reduced CENP-E levels at KTs by ~40% (*Figure 7D and E*, right panels), in agreement with previous reports (*Johnson et al., 2004*; *Sharp-Baker and Chen, 2001*). Taken together, these results show that Bub1 kinase activity is largely dispensable for chromosome congression and segregation. It follows that even though Bub1 inhibition results in a marked reduction of Aurora B levels at centromeres (*Figure 4*), these levels are still sufficient to ensure largely faithful chromosome segregation. Conversely, Bub1 protein is clearly important for efficient chromosome congression, presumably reflecting the role of Bub1 in CENP-E recruitment to KTs.

## Bub1 inhibition sensitizes HeLa cells to clinically relevant doses of Paclitaxel

Interference with the SAC proteins Mps1 or BubR1 was previously shown to exert synergistic effects with Paclitaxel treatment of tumor cells, significantly elevating the frequency of chromosome missegregation and lethality (*Janssen et al., 2009*; *Lee et al., 2004*). Thus, we asked how inhibition of Bub1 kinase activity by BAY-320 or BAY-524 would impact on cells in which MT dynamics was

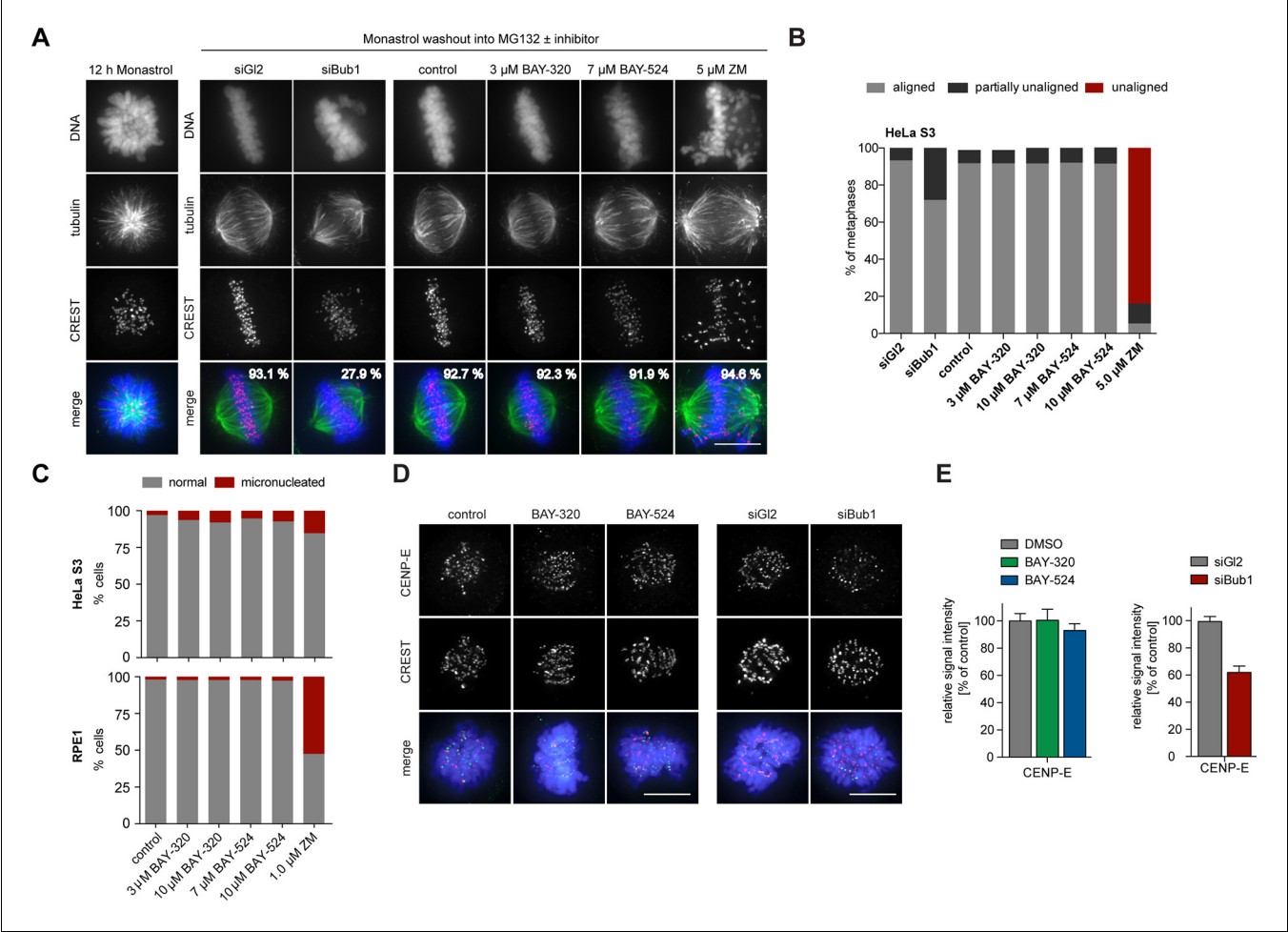

**Figure 7.** Bub1 inhibition does not significantly affect chromosome congression. (A) HeLa S3 cells were transfected with control (Gl2) or Bub1 siRNA-oligonucleotides for 48 hr, synchronized by thymidine block and released for 12 hr in the presence of the Eg5 inhibitor monastrol to induce the formation of monopolar spindles. The capacity of spindle bipolarization and metaphase plate formation was tested by monastrol wash-out and addition of MG132 and indicated drugs for 2 hr (n = 170–200 cells). Percentages indicate the frequencies of depicted spindle morphologies. (B) Histograms show the frequencies of full, partial (≤5 unaligned chromosomes) or failed metaphase chromosome alignments that were observed in the experiment shown in (A). (C) HeLa S3 and RPE1 cells were treated for 16 hr with the indicated drugs, fixed and analyzed by IFM. Histograms show the frequency of micronucleation among interphase cells (n = 300 cells per condition). (D) Depletion but not inhibition of Bub1 kinase affects recruitment of CENP-E to unattached kinetochores. Untreated HeLa S3 cells or cells transfected with control (Gl2) or Bub1 siRNA-oligonucleotides (for 48 hr) were synchronized by thymidine block and released for 10 hr in the presence or absence of 3 μM BAY-320 or 7 μM BAY-524. The cells were fixed and stained for CENP-E, CREST, DNA (DAPI) and analyzed by IFM. (E) Histograms show average CENP-E KT levels observed in prometaphase cells. Data relate to micrographs shown in (D). Error bars represent SEM. Scale bars represent 10 μm.

compromised by low doses of Paclitaxel. Importantly, when used at clinically relevant doses of 1–4 nM, Paclitaxel induces spindle defects and aneuploidy without delaying mitotic progression (*Brito and Rieder, 2009*; *Chen and Horwitz, 2002*; *Ikui et al., 2005*; *Janssen et al., 2009*). While single treatment with 1–4 nM Paclitaxel produced modest impairment of cell proliferation, the concomitant application of the Bub1 inhibitors, BAY-320 at 3 μM or BAY-524 at 7 or 10 μM, clearly exacerbated inhibition of proliferation. Effects were particularly drastic in aneuploid HeLa cells (*Figure 8A and B*, top panels), while diploid RPE1 cells were less affected (*Figure 8A and B*, bottom panels). For comparison, we also examined the effects of combining low dose Paclitaxel treatment with partial inhibition of Mps1 by Reversine (*Janssen et al., 2009*). This analysis shows that the combination of Paclitaxel with either Mps1 or Bub1 inhibition produced similar synergistic effects, albeit with cell-type specific differences (*Figure 8A and B*). Using extensive dose-response analyses, synergy between BAY-320 and Paclitaxel treatment was further confirmed for both HeLa (*Figure 8C*)

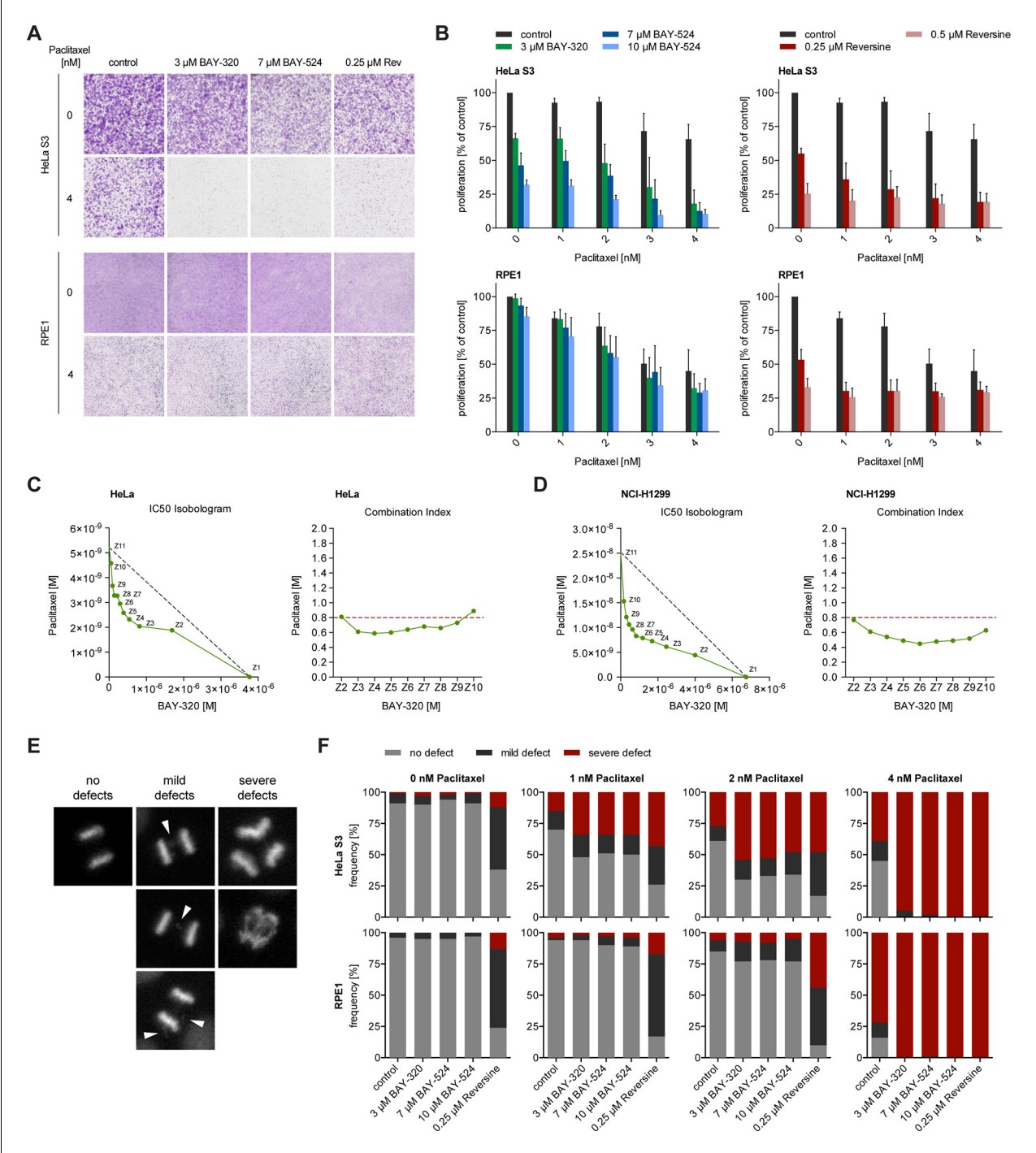

**Figure 8.** BAY-320 and BAY-524 treatment sensitizes cells to low doses of Paclitaxel. (A) Micrographs show colony formation of HeLa (top panel) and RPE1 cells (bottom panel) treated for 7 days with solvent (control) or the indicated kinase inhibitors in the presence or absence of 4 nM Paclitaxel. (B) Histograms quantify colony formation in HeLa (top panels) and RPE1 cells (bottom panels) treated with the indicated kinase inhibitors in the presence or absence of 1–4 nM Paclitaxel for 7 days. (C, D) $IC_{50}$-Isobolograms confirm the synergistic effect of BAY-320 and Paclitaxel on cell survival. HeLa cells (C) or NCI-H1299 non-small cell lung cancer cells (D) were grown in the presence various concentrations of BAY-320 (0.1–10 μM) and paclitaxel (1–100 nM) in mono (Z1, Z11) and in nine different fixed-ratio combinations (Z2-Z10). $IC_{50}$ values were determined and the respective BAY-320 and Paclitaxel concentrations plotted in $IC_{50}$ Isobolograms (left panel). The grey dashed lines indicate the results expected for additivity. Combination indices (CIs) were calculated according to the median-effect model of Chou-Talalay (**Chou, 2006**) and plotted over fixed-ratio combinations Z2-Z10 (right panel). The red dashed line indicates a CI of 0.8 (defined as upper limit for a synergistic interaction). (E) Time-lapse stills of HeLa cells expressing H2B-GFP illustrate chromosome segregation defects that were used to classify cell fates in the experiments described in (F); arrowheads point to chromosome

*Figure 8 continued on next page*

*Figure 8 continued*

bridges and lagging chromosomes. (F) HeLa (top panels) and RPE1 cells (bottom panels) stably expressing H2B-GFP were treated with solvent (control) or the indicated kinase inhibitors in the presence or absence of 1–4 nM Paclitaxel and monitored by fluorescence time-lapse imaging. Histograms show the frequencies of chromosome segregation defects, following the classification illustrated in (E) (n = 100 cells per condition).

and non-small cell lung cancer cells (*Figure 8D*). In future, it will be interesting to determine to what extent combined treatments differentially affect aneuploid *versus* diploid cells (*Janssen et al., 2009*; *Kops et al., 2004*; *Maia et al., 2015*).

To assess whether the observed impairment of proliferation results from errors in chromosome segregation (*Kops et al., 2004*), we scored HeLa and RPE1 cells expressing GFP-H2B for mild and severe chromosomal defects, as illustrated in *Figure 8E*. Following application of the above Paclitaxel and Bub1/Mps1 inhibitor treatments, the frequencies of chromosomal defects were monitored by fluorescence time-lapse imaging and quantified (*Figure 8F*). Consistent with the micronucleation data described above (*Figure 7*), Bub1 inhibition alone did not significantly elevate the frequency of chromosome missegregation in either HeLa or RPE1 cells (*Figure 8F*). For comparison, interference with the SAC by inhibition of Mps1 led to a marked increase in segregation defects in both cell lines, as expected (*Figure 8F*). Most importantly, HeLa cells displayed an even higher frequency of severe chromosome segregation defects when Bub1 inhibition was combined with 1–4 nM Paclitaxel, comparable to the consequences of combined Mps1 inhibition and Paclitaxel treatment (*Figure 8F*). In contrast, Bub1 inhibition only marginally elevated the rate of Paclitaxel-induced chromosome missegregation in RPE1 cells, while combinatorial treatment with Reversine still resulted in a high rate of mild segregation defects. Considering the strong correlation in the data shown in *Figures 7* and *8*, it is tempting to conclude that chromosome segregation errors constitute the most likely cause for the observed impairment of cell proliferation when Bub1 inhibition is combined with low dose Paclitaxel treatment. Thus, although inhibition of Bub1 kinase activity per se exerts only minor effects on SAC functionality, chromosome segregation and mitotic progression, treatment with BAY-320 or BAY-524 sensitizes cells to low, clinically relevant doses of Paclitaxel. These findings are clearly relevant for the potential therapeutic use of Bub1 inhibitors.

## Discussion

Bub1 kinase is important for chromosome congression and the fidelity of chromosome segregation in species from yeast to vertebrates (reviewed in [*Elowe, 2011*; *Funabiki and Wynne, 2013*]). However, the role of the catalytic activity of Bub1 in different species remained a matter of debate. Here, we have characterized two novel small molecule inhibitors of Bub1, BAY-320 and BAY-524, and used these reagents to explore the role of Bub1 catalytic activity in mitotic processes. A systematic comparison of the phenotypic consequences of Bub1 inhibition with those of siRNA-mediated Bub1 depletion leads us to conclude that Bub1 functions primarily as a scaffolding protein. However, we see prominent effects of Bub1 kinase inhibition on chromosome arm cohesion and CPC localization and subtle effects on mitotic progression, including the efficacy of the SAC. Finally, we demonstrate a striking synergy between inhibition of Bub1 activity and Paclitaxel-induced interference with MT dynamics, which manifests itself in a marked increase in chromosome segregation errors and drastically reduced cell proliferation. Our results contribute to clarify the role of Bub1 kinase activity in different mitotic processes. Moreover, they have important implications for potential use of Bub1 inhibitors in therapeutic applications.

### Validation of BAY-320 and BAY-524 as Bub1 inhibitors

Clarification of the role of Bub1 activity in mitosis has previously been hampered by the absence of specific inhibitors. While genetic or siRNA-mediated depletion experiments generally suffer from poor temporal resolution, small molecule inhibitors offer a unique opportunity for acute kinase inactivation. The only previously described inhibitor of Bub1 is the bulky ATP analog 2OH-BNPP1 (*Kang et al., 2008*; *Krenn et al., 2012*; *Liu et al., 2015*; *Nyati et al., 2015*), but neither the specificity nor the efficacy of this compound in intact cells have been thoroughly characterized. Here, we document that both BAY-320 and BAY-524 effectively inhibit Bub1 kinase activity, in intact cells as

well as in vitro. In addition to inhibiting phosphorylation of histone H2A-T120, both compounds cause a persistence of chromosome arm cohesion, validating their efficacy (*Kawashima et al., 2010*; *Liu et al., 2015*). Moreover, in vitro assays performed on a large panel of kinases showed that inhibition of off-target substrates required at least 20x higher concentrations of BAY-320 than inhibition of Bub1, and for one potentially relevant off-target, the kinase Haspin, we show that intracellular phosphorylation of a major substrate of this kinase is not decreased by inhibitor concentrations that effectively inhibit Bub1. Furthermore, a binding assay performed on a panel of 403 human kinases documents exquisite selectivity of BAY-320 for Bub1. Thus, we are confident that these new Bub1 inhibitors constitute highly effective and specific tools to explore the role of Bub1 kinase activity.

## Impact of Bub1 inhibition on mitotic progression

The major conclusion emerging from the present study is that the overall impact of Bub1 inhibition on mitotic progression is surprisingly mild, clearly less severe than the impact of Bub1 depletion. This reinforces the notion that the requirement for Bub1 during chromosome congression and segregation primarily reflects a scaffolding function (*Fernius and Hardwick, 2007*; *Klebig et al., 2009*; *Perera and Taylor, 2010b*; *Rischitor et al., 2007*; *Sharp-Baker and Chen, 2001*). It is difficult to exclude that a small fraction of total Bub1 kinase activity might be refractory to inhibition and suffice for functionality, but we emphasize that genetic elimination of Bub1 kinase activity is compatible with mouse development, arguing similarly against an essential role of Bub1 kinase activity for mitotic progression (*Ricke et al., 2012*). Thus, inhibitor studies and genetic data concur to indicate that lack of Bub1 kinase activity produces only mild disturbances of mitotic progression.

We show that Bub1 inhibition by BAY-320 or BAY-524 results in loss of Shugoshin and CPC subunits proteins from centromeres/KTs, and that CPC levels at these locations are further reduced upon simultaneous inhibition of Bub1 and Haspin, in line with the established roles of these kinases in the phosphorylation of histones H2A-T120 and H3-T3, respectively (*Wang et al., 2011*; *Yamagishi et al., 2010*). Furthermore, both Sgo2 and Aurora B relocalize to chromosome arms when Bub1 is inhibited. Phosphorylation of Sgo2 by Aurora B promotes Sgo2 interaction with phosphatase 2A (PP2A) (*Tanno et al., 2010*), and PP2A in turn protects cohesin proteins against phosphorylation (*Kitajima et al., 2006*; *Riedel et al., 2006*). Thus, colocalization of Sgo2 and Aurora B in Bub1-inhibited cells provides a straightforward explanation for the observed persistence of arm cohesion (*Kitajima et al., 2005*; *Perera et al., 2007*). However, alternative mechanisms should not be excluded and it might be rewarding to explore a possible connection between Bub1 kinase and the Sororin-Wapl pathway (*Peters and Nishiyama, 2012*).

It may appear surprising that the persistent chromosome arm cohesion observed in Bub1 inhibited cells did not markedly prolong mitotic timing (*Figure 2*). However, we note that depletion of Wapl also causes persistent cohesion without significantly affecting mitotic progression (*Lara-Gonzalez and Taylor, 2012*). In contrast to Bub1 inhibition, depletion of Bub1 markedly extended mitotic timing (*Figure 2*). One straightforward explanation for this observation is that Bub1 depletion, but not Bub1 inhibition, caused the displacement of CENP-E from KTs, a motor protein required for efficient chromosome congression (*Bancroft et al., 2015*; *Putkey et al., 2002*; *Tanudji et al., 2004*). A mechanism centered on CENP-E may also explain the observation that Bub1 depletion exerted a more extensive mitotic delay in the hypertriploid HeLa cells than in diploid RPE1 cells.

Considering the important role of Aurora B in the regulation of KT-MT interactions (*Carmena et al., 2012*; *Funabiki and Wynne, 2013*), it is remarkable that the observed reduction of centromere/KT-associated CPC caused by Bub1 inhibition did not exert a more profound effect on the fidelity of chromosome segregation. This suggests that approximately half the normal centromere/KT levels of CPC are sufficient to confer functionality. In line with this conclusion, we note that partial impairment of CPC recruitment to centromeres/KTs did not abolish viability or trigger extensive defects in chromosome segregation in budding yeast or chicken DT40 cells (*Campbell and Desai, 2013*; *Yue et al., 2008*).

Inhibition of Bub1 kinase activity did not significantly reduce the KT recruitment of Mad1, Mad2 and BubR1 and barely affected the ability of nocodazole-treated cells to maintain a SAC arrest. Even when SAC activity was compromised by partial inhibition of the SAC kinase Mps1, Bub1 inhibition triggered only minor weakening of SAC signaling. In striking contrast, Bub1 depletion produced a drastic weakening of the SAC in this sensitized background. For comparison, combined inhibition of

Mps1 and either Plk1 or Aurora B resulted in a complete SAC shutdown and immediate mitotic exit, in line with previous results (*Saurin et al., 2011*; *von Schubert et al., 2015*). Collectively these findings confirm that mitotic functions of Bub1 depend primarily on Bub1 protein rather than kinase activity. In future, it will be interesting to explore whether Bub1 activity contributes to purported non-mitotic functions of Bub1 (*Nyati et al., 2015*; *Yang et al., 2012*).

### Use of Bub1 inhibitors for therapeutic applications

Inhibition of SAC kinases has emerged as a potentially attractive strategy to kill tumor cells (*Janssen et al., 2009*; *Salmela and Kallio, 2013*). Several inhibitors of the SAC kinase Mps1 were shown to exert anti-tumor effects in mouse models (*Colombo et al., 2010*; *Kusakabe et al., 2015*; *Tannous et al., 2013*; *Tardif et al., 2011*), but toxicity associated with single agent therapy remains a concern (*Martinez et al., 2015*). Instead, combination of anti-SAC compounds with MT-targeting agents may constitute a more rewarding strategy (*Jemaà et al., 2013*; *Maia et al., 2015*). Our present data suggest that it may be attractive to use inhibitors of Bub1 in combinatorial therapy. While BAY-320 and BAY-524 had comparatively little effect on mitotic progression when used as single agents, they showed extensive anti-proliferative activity, accompanied by strong increases in chromosome segregation errors, when combined with therapeutic doses of Paclitaxel. A plausible explanation for this synergy is that Paclitaxel increases KT-MT attachment errors to levels that can no longer be corrected when Aurora B/CPC is partially displaced upon Bub1 inhibition. Interestingly, these synergistic effects were substantially more pronounced in aneuploid HeLa cells than in near-diploid RPE1 cells, suggesting a potential therapeutic window. These findings clearly encourage further exploration of the potential use of Bub1 inhibitors for therapeutic applications.

## Materials and methods

### Preparation of BAY-320 and BAY-524 inhibitors

BAY-320 and BAY-524 were synthesized as described previously (*Hitchcock et al., 2013*). For biochemical and cellular experiments BAY-320 and BAY-524 were used from stock solutions in dimethyl sulfoxide (DMSO). Working concentration of Bub1 inhibitors are indicated in Figures and Figure legends, respectively.

### Determination of $IC_{50}$-concentrations

Inhibitory activities BAY-320 and BAY-524 towards Bub1 in presence of 2 mM ATP were quantified as previously published (*Hitchcock et al., 2013*). A time-resolved fluorescence energy transfer (TR-FRET) kinase assay was used to measure phosphorylation of the synthetic peptide Biotin-Ahx-VLLPK-KSFAEPG (C-terminus in amide form, Biosyntan, Berlin, Germany) by the recombinant catalytic domain of human Bub1 (amino acids 704–1085). Recombinant human Bub1 (704–1085) was expressed in Hi5 insect cells with an N-terminal His6-tag and purified by affinity- (Ni-NTA) and size exclusion chromatography.

### Kinase selectivity profiling

BAY-320 was screenedin vitro, at 10 µM and 10 µM ATP, against a panel of 222 kinases using the Eurofins kinase profiler screen (Millipore). In addition, BAY-320 was screened, at 300 and 1000 nM, in an active site-directed competition-binding assay measuring 403 human kinases (Lead Hunter, DiscoverX Kinome Scan).

### In vitro kinase assay

HEK 293T cells were transfected with plasmids coding for LAP-tagged Bub1 wild-type (WT) or the K821R kinase-dead (KD) mutant (kindly provided by G. Kops, Utrecht, Netherlands) (*Suijkerbuijk et al., 2012*). After induction of mitotic arrest (18 hr incubation with 1 µg/ml of nocodazole), the cells were harvested and lysed in kinase lysis buffer (50 mM HEPES pH7.5, 150 mM NaCl, 5 mM EDTA, 0.5% NP-40, 1 mM $Na_3VO_4$, 1 mM β-glycerophosphate, 1 mM NaF and complete protease inhibitor (Roche)). Lysates were cleared by centrifugation for 15 min at 21,000 *g*, 4°C, and LAP-Bub1 proteins isolated by a 2 hr incubation with S-protein-agarose (Novagen, EMD Chemical, CA, USA). Beads were washed six times in lysis buffer containing increasing concentrations of NaCl

(150 mM, 200 mM, 300 mM, 400 mM, 500 mM and 600 mM) and three times in kinase buffer (20 mM HEPES pH7.5, 100 mM KCl, 10 mM MgCl, 1 mM Na3VO4, 1 mM -glycerophosphate, 1 mM NaF, 1 mM DTT). The bead-bound LAP-Bub1 was then aliquoted and used for kinase assays in 30-µl reaction volumes. Kinase reactions were carried out at 30°C in kinase buffer in the presence of 100 µM ATP, 5 µCi γ-$^{32}$P-ATP, 1 µg recombinant histone H2A (NEB, Frankfurt am Main, Germany) as substrate, and serial dilutions of Bub1 inhibitors. The reactions were stopped after 30 min by the addition of sample buffer and heating to 95°C. The samples were then resolved by SDS-PAGE and visualized by autoradiography and Western blotting.

## Cell culture

HeLa S3 cells, HeLa S3 cells expressing histone H2B-GFP (*Silljé et al., 2006*), HeLa Kyoto cells expressing a FRET reporter for Aurora B fused to histone H2B (*van der Waal et al., 2012*) and HEK293T cells were grown under standard conditions in DMEM-Glutamax medium (Invitrogen, CA, USA), supplemented with 10% heat-inactivated fetal calf serum (FCS) (PAN Biotech, Aidenbach, Germany) and penicillin-streptomycin (Pen-Strep; 100 IU/ml and 100 mg/ml respectively, Gibco Life Technologies, Zug, Switzerland). hTERT-RPE1 cells and hTERT-RPE1 cells expressing histone H2B-GFP (kind gift of Stephen Taylor, University of Manchester, UK) were cultured in F12 DMEM nutrient mixture F-12 HAM (Sigma Aldrich, MO, USA) supplemented with 10% heat-inactivated FCS, L-glutamine (2 mM; PAN Biotech, Aidenbach, Germany), sodium bicarbonate (0.35%; Sigma-Aldrich, MO, USA) and Pen-Strep. NCI-H1299 cells were grown under standard conditions in RPMI-1640 medium supplemented with L-glutamine (Biochrome, Berlin, Germany) and 10% fetal calf serum (Biochrome, Berlin, Germany). All cell lines were routinely tested for mycoplasma, using PCR (by the lab in Basel) or the MycoAlert Mycoplasma Detection Assay (by the lab in Berlin). HeLa cells (ACC-57) were obtained from the German Collection of Microorganisms and Cell Cultures, Braunschweig, and authentication was done at provider prior to shipment; NCI-H1299 (CRL-5803) were obtained from ATCC and authentication was done by STR profiling (authentication service at German Collection of Microorganisms and Cell Cultures, Braunschweig). Thymidine arrest was performed for 24 hr and cells were either released into fresh medium for 10 hr or into medium supplemented with Nocodazole for 12–14 hr. Thymidine (Sigma-Aldrich) was used at 2 mM if not stated otherwise, Nocodazole (Sigma-Aldrich) at 3.3 µM if not stated otherwise, RO-3306 at 10 µM (Calbiochem, Darmstadt, Germany), Paclitaxel (Calbiochem) at 1–4 nM, Reversine (Enzo Life Sciences, Lausen, Switzerland) at 0.25 and 0.5 µM, ZM-447439 (Tocris Bioscience, [*Ditchfield et al., 2003*]) at 1.25, 2.5 and 5.0 µM, 5-Iodotubercidin (5'Itu, Santa Cruz Biotechnology, TX, US) at 2.5 µM, Monastrol (Enzo Life Sciences) at 150 µM and MG132 (Calbiochem) at 10 and 20 µM.

## Transient plasmid transfection and siRNA-mediated protein depletion

Transient transfections of HEK293T cells with plasmids and small interfering RNA (siRNA) duplexes were performed using TransIT-LT1 transfection reagent (Mirus Bio, Madison, WI) and Oligofectamine (Invitrogen), respectively, according to manufacturers protocols. The following siRNA duplex oligonucleotides were used: siGl2 CGTACGCGGAATACTTCGA (*Elbashir et al., 2001*), siBub1 CCAGGCTGAACCCAGAGAGTT (Tang 2004). All siRNA duplex oligonucleotides were ordered from Qiagen (Hilden, Germany).

## Fluorescence-activated cell sorting

HeLa S3 or RPE1 cells were incubated with kinase inhibitors or depleted of the indicated proteins for 48 hr. Cell suspensions were then fixed with 70% ice-cold ethanol and incubated with 0.2 mg/ml RNase (Sigma-Aldrich) and 5 µg/ml propidium iodide (Sigma-Aldrich). Cellular DNA content was determined by flow cytometry using FACSCanto II (BD Biosciences Clontech, San Jose, CA, USA) and FlowJo (Treestar, Ashland, OR, USA) instruments.

## Cell extracts and sample preparation for Western blot analysis

Cells extracts were prepared on ice for 30 min in Tris lysis buffer (20 mM Tris, pH 7.4, 150 mM NaCl, 0.5% IGEPAL CA-630, 30 µg/ml RNAse, 30 µg/ml DNAse, 1 mM DTT, protease inhibitors cocktail (Roche, Basel, Switzerland) and phosphatase inhibitor cocktails (cocktails 2 and 3, Sigma-Aldrich).

Lysates were cleared by centrifugation for 15 min at 21,000 g, 4°C, and proteins were resolved by SDS-PAGE and analyzed by Western blotting.

## Histone isolation

HeLa S3 cells were as described above and mitotic cells were collected by shake-off. The cells were the washed with cold PBS and lysed at 4°C for 30 min using histone lysis buffer (50 mM Tris pH 7.8, 300 mM NaCl, 1% IGEPAL CA-630). Nuclei were collected by centrifugation (110 g, 4°C, 10 min) and washed three times with histone lysis buffer. After an additional wash with Tris-EDTA (100 mM Tris, 1 mM EDTA), the nuclear pellet was incubated for 2 hr in 0.4 M HCl at 4°C. After high-speed centrifugation of the sample, 6 volumes of acetone were added to the supernatant, followed by overnight incubation at -20°C. Histones were collected by centrifugation, washed with acetone, air-dried and resolved by SDS-PAGE.

## Antibodies

Antibodies used for Western blotting: anti-Bub1 ([*Hanisch et al., 2006*] or ab9000, Abcam, Cambridge, UK), anti-pT120-H2A (Active Motif, Carlsbad, CA, USA) and anti-α-tubulin (DM1A, Sigma-Aldrich). Antibodies used for immunofluorescence microscopy: anti-Mad1 (clone 117–468 [*Fava et al., 2011*]), anti-cMad2 (clone 107–276 [*Fava et al., 2011*]) anti-Borealin (*Klein et al., 2006*), anti-INCENP (clone 58–217, ab23956, Abcam), anti-Bub1 (antibody against Bub1 hybridoma (clone 62–406) was produced after mice were injected with Bub1 recombinant protein spanning residues 1–318, anti-Bub1 (ab9000, Abcam), CREST anti-human auto-immune serum (Immunovision, Springdale, AR, USA), anti-Aurora B (AIM-1, BD Biosciences, San Jose, CA, USA), anti-Bub1 (ab9000, Abcam), anti-CENP-E (1H12, Abcam), anti-Mad2 (A300-301A, Bethyl Laboratories, Montgomery, TX, USA), anti-Sgo1 (Abnova, Taipei, Taiwan), anti-Sgo2 (Bethyl Laboratories), anti-pT120-H2A (Active Motif, Carlsbad, CA, USA; ABIN482721), anti-pS7CENP-A (clone NL41, Merck Millipore, Billerica, MA, USA), anti-pT3-H3 (clone 9714, Cell Signaling Technology, Danvers, MA, USA) and anti-pS10-H3 (Millipore, Billerica, MA, USA). The polyclonal MCAK (R120) antibody was raised in rabbits by immunization with bacterially expressed His-MCAK$_{aa588-725}$. For immunofluorescence experiments, all primary antibodies were detected with AlexaRed-594-, AlexaRed-564-, and AlexaGreen-488-labeled secondary anti-mouse and anti-rabbit antibodies (Invitrogen, Carlsbad, CA, USA) or Cy5-conjugated donkey antibodies (Dianova, Hamburg, Germany). For Western blotting, signals were detected using HRP-conjugated anti-mouse or anti-rabbit antibodies (Pierce, Rockford, IL, USA).

## In-cell western

HeLa cells were seeded into 96-well plates for 5 hr at 37°C (ca. 25'000 cells/well). Then, cells were treated with Nocodazole for 16 hr and varying concentrations of test compounds for 1 hr. The cells were fixed, washed and blocked with buffer before incubating with the primary antibody (Phospho-Histone H2A; ABIN482721; 1:200) overnight at 2–8°C. After washing, secondary IRDye-labeled antibody mix with cell stains was added for 1 hr and washed again. Plates were scanned with a LiCor Odyssey Infrared Imager at 800 nm for P-H2A and at 700 nm for Draq5/Sapphire, a cell stain. The signal ratio (800/700 nm) for cells treated only with Nocodazole was set to 100% and the corresponding ratio for untreated cells to 0%. The IC$_{50}$ value was then determined by curve fitting (using a four parameter fit).

## Immunofluorescence microscopy, image processing, quantification and live cell imaging

For fluorescence microscopy cells were grown on coverslips and fixed in PTEMF buffer (20 mM PIPES, pH 6.8, 0.2% Triton X-100, 10 mM EGTA, 1 mM MgCl$_2$, 4% formaldehyde) or methanol at -20°C (for CENP-A pS7), respectively. Images of randomly selected cell were acquired as z-stacks using a DeltaVision microscope (GE Healthcare) on an Olympus IX71 base (Applied Precision, WA, USA), equipped with a Plan Apochromat N 60x/NA1.42 oil immersion objective (Olympus) and a CoolSNAP HQ2 camera (Photometrics). Serial optical sections were deconvolved and projected using SoftWorx software (GE Healthcare). Images were quantified as previously described (*von Schubert et al., 2015*) using automated pipelines run by Cell Profiler software (*Carpenter et al., 2006*). Results from 2–3 independent experiments were pooled and statistical

analysis was done with GraphPad Prism software. Error bars on histograms illustrate SEM. Scale bars represent 10 µm.

For time-lapse imaging, cells were imaged using a Nikon ECLIPSE Ti microscope equipped with a CoolLED pE-1 illumination system and a 20x/NA0.75 air Plan Apochromat objective (Nikon) in a climate-controlled environment. Images were acquired at multiple positions at indicated time intervals. MetaMorph 7.7 software (MDS Analytical Technologies, Sunnyvale, CA, USA) was used for acquisition and processing of data. FRET, FRAP, and high sensitivity microscopy (monitoring endogenously EGFP-tagged proteins) experiments were carried out using a spinning disk confocal system (Intelligent Imaging Innovations) based on a Zeiss Axio Observer stand equipped with a Photometric Evolve 512 back-illuminated EMCCD camera, 63x/NA1.4 plan apochromat objective and diode lasers and run by SlideBook software. FRET analyses were carried out by excitation with a 440 nm diode laser and by recording of CFP (CFP signal) and YFP (FRET signal) fluorescence emission in z-stacks. Background-corrected FRET ratios (CFP signal/FRET signal) were calculated in ImageJ using the Ratio Plus plugin. FRAP analysis of EGFP-Bub1 was performed with a 488 diode laser on one KT pair per cell. Overall bleaching was corrected using the signal intensities at a cytoplasmic region not targeted for photobleaching (average of the first 4 frames). Fluorescence recovery half-times and plateaus were determined by non-linear curve fitting based on a one-phase association in Prism software (GraphPad).

## Colony formation assay

Asynchronous cell cultures (50,000/well) were plated on 6-well plates (Falcon). After 7 days of proliferation in the presence of the indicated drugs, the cells were fixed with ice-cold methanol at -20°C and stained with 0.1% Cresyl Violet according to standard procedures. Dried culture plates were scanned and intensities measured using ImageJ after black-and-white-conversion and inversion of the images.

## $IC_{50}$ isobolograms

BAY-320 plus Paclitaxel combination studies were conducted with HeLa and NCI-H1299 cells. Cells were plated into 384-well plates at 600 (HeLa) or 700 (NCI-H1299) cells per well. After 24 hr, cells were treated with BAY-320 (concentration range, 1E-07 M to 1E-05 M) and Paclitaxel (concentration range, 1E-09 M to 1E-07 M) for single compound treatments, and in nine different fixed-ratio combinations of BAY-320 (D1) and Paclitaxel (D2) (0.9xD1+0.1xD2, 0.8xD1+0.2xD2, 0.7xD1+0.3xD2, 0.6xD1+0.4xD2, 0.5xD1+0.5xD2, 0.4xD1+0.6xD2, 0.3xD1+0.7xD2, 0.2xD1+0.8xD2, 0.1xD1 +0.9xD2). Cell viability was assessed after 96 hr exposure, using the Cell Titre-Glo Luminescent Cell Viability Assay (Promega). $IC_{50}$ values were determined by means of a 4-parameter fit after normalization of measurement data to vehicle (DMSO)-treated cells (=100%) and readings taken immediately before compound addition (=0%). $IC_{50}$ isobolograms were plotted with the actual concentrations of the two compounds on the x- and y-axis, and the combination index (CI) was calculated according to the median-effect model of Chou-Talalay (*Chou, 2006*). A CI of $\leq 0.8$ was defined as more than additive (i.e. synergistic) interaction, and a CI of $\geq 1.2$ was defined as antagonistic interaction.

## rAAV-mediated gene targeting

For gene targeting, homology arms to human Bub1 (*BUB1*) gene were amplified from RPE1 cell genomic DNA. Targeting constructs allowing the insertion of an EGFP tag C-terminal to Bub1 were assembled by 4-piece ligation in a *Not*I-digested pAAV vector. Recombinant adenovirus-associated virus (rAAV) particles were generated as previously described (*Berdougo et al., 2009*). RPE1 cells were infected with 3 ml of viral supernatant for 48 hr and then expanded into fresh medium for an additional 48 hr. FACS sorting was used to select EGFP-positive cells, as previously described (*Collin et al., 2013*). To facilitate detection of fluorescence at mitotic stages, cells were synchronized with RO-3306 (10 µM) for 18 hr and released into nocodazole (50 nM) for 2 hr, before they were trypsinized and subjected to sorting in the continued presence of nocodazole (10 nM). Infected or uninfected cells were filtered (30 µm, Partec) and EGFP-positive cells (488 excitation, 514/30 emission filter) were isolated on an Aria IIIu (BD) cell sorter by selecting the 514/30 channel against a

585/42 filter detecting cellular autofluorescence. Single cells were sorted into 96-well plates filled with conditioned medium and positive clones screened for by fluorescence microscopy.

## Acknowledgements

We thank the project team at Bayer Pharma AG for their contribution. We also thank Prof. GJPL Kops (University Medical Center Utrecht) and Dr. M Vleugel (University of Technology Delft) for providing the LAP-Bub1 and LAP-Bub1(KD) plasmids, Stephen Taylor (University of Manchester) for the RPE1 H2B-GFP cell line, Zhen Dou (University of Science and Technology of China, Hefei) for the generation of the MCAK antibody and Michael Lampson (University of Pennsylvania) as well as Daniel Gerlich (Institute of Molecular Biotechnology of the Austrian Academy of Sciences, Vienna) for Aurora B biosensor constructs and HeLa Kyoto cell lines. We also thank the Biozentrum's Imaging Core Facility for support (O Biehlmaier, A Loynton-Ferrand and N Ehrenfeuchter) and Janine Zankl (Biozentrum FACS facility) for help with FACS analysis. APB was funded by a fellowship from the 'Fellowships for Excellence' International PhD Program (University of Basel, Switzerland) supported by the Werner von Siemens Foundation. This work was supported by the University of Basel and the Swiss National Science Foundation (310030B_149641).

## Additional information

### Competing interests

GS: Employee of Bayer Pharma AG. MH: Employee of Bayer Pharma AG. AM: Employee of Bayer Pharma AG. JS: Employee of Bayer Pharma AG. AFM: Employee of Bayer Pharma AG. FvN: Employee of Bayer Pharma AG. DM: Employee of Bayer Pharma AG. The other authors declare that no competing interests exist.

### Funding

| Funder | Grant reference number | Author |
| --- | --- | --- |
| Fellowships for Excellence International PhD Program, Werner von Siemens Foundation | Graduate Student Fellowship | Anna Pauline Baron |
| Swiss National Science Foundation | 310030B_149641 | Anna Pauline Baron Conrad von Schubert Erich A Nigg |
| Universität Basel | | Anna Pauline Baron Conrad von Schubert Erich A Nigg |

The funders had no role in study design, data collection and interpretation, or the decision to submit the work for publication.

### Author contributions

APB, Conception and design, Acquisition of data, Analysis and interpretation of data, Drafting or revising the article; CvS, Conception and design, Acquisition of data, Analysis and interpretation of data, Drafting or revising the article; FC, MH, AM, JS, AFM, FvN, Acquisition of data, Analysis and interpretation of data; GS, Conception and design, Analysis and interpretation of data; DM, Conception and design, Acquisition of data, Analysis and interpretation of data; EAN, designed the experiments; wrote the manuscript with help from all co-authors, Conception and design, Analysis and interpretation of data, Drafting or revising the article

### Author ORCIDs

Erich A Nigg, http://orcid.org/0000-0003-4835-5719

# Additional files

**Supplementary files**

• Supplementary file 1. Related to *Figure 1*. IC$_{50}$ profile of BAY-320 and BAY-524.

• Supplementary file 2. Related to *Figure 1*. In vitro kinase-selectivity profile of BAY-320 on a panel of 222 human kinases (Eurofins kinase profiler screen, Millipore). Shown are percentages of residual kinase activity at 10 μM BAY-320 and 10 μM ATP.

• Supplementary file 3. Related to *Figure 1*. In vitro kinase-selectivity profile of BAY-320. Results are based on an active site-directed competition-binding assay using a panel of 403 human kinases (DiscoverX Kinome Scan, Lead Hunter). Results show ligand binding at 300 and 1000 nM BAY-320 relative to control condition.

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
