## [Decision Letter]

Thank you for submitting your work entitled "Probing the catalytic functions of Bub1 kinase using the small molecule inhibitors BAY-320 and BAY-524" for consideration by *eLife*. Your article has been reviewed by three peer reviewers, one of whom is a member of our Board of Reviewing Editors (Jonathon Pines). The evaluation has been overseen by the Reviewing Editor and Ivan Dikic as the Senior Editor.

The reviewers have discussed the reviews with one another and the Reviewing Editor has drafted this decision to help you prepare a revised submission.

Summary:

In this study the authors have developed small molecule inhibitors that specifically inhibit the Bub1 kinase. They demonstrate the specificity of their inhibitors in vitro on a panel of kinases, and in vivo using phosphorylation of Histone H2A threonine 120 as a read out. They show that inhibiting Bub1 causes mild effects on the timing of mitosis and on chromosome congression, but causes persistent cohesion of chromosomes arms concomitant with the chromosome passenger complex remaining on the arms. Although the effects of the Bub1 kinase inhibitors alone only mildly affect mitosis, they have a synergistic effect with low doses of taxol.

Essential revisions:

All three reviewers recognised the quality of the study, and that specific Bub1 kinase inhibitors would be very useful research tools. There were two major experimental concerns, however:

1) More evidence for the specificity of the compounds should be presented. At present this mostly consists of a patent and kinase assays on a panel of kinases.

2) Are the effects of the compounds in cells specific to inhibiting Bub1? The authors are using a relatively high concentration of the inhibitors (3 and 7 μM for BAY-320 and BAY-524, respectively), and the toxic concentration is 10 μM. Moreover, there is still some residual H2A-T120 phosphorylation (Figure 1), which suggests Bub1 activity is not maximally inhibited under these conditions.

Please address these concerns by providing further experimental data. For example, what are the phenotypes of cells treated with the inhibitors in which Bub1 has been depleted and rescued with a kinase-dead construct? If the inhibitors are specific to Bub1 there should be no further effect of the inhibitors.

In addition to this experimental concern, the reviewers suggest some re-writing is required to clarify what is currently known about the function of Bub1 kinase activity. One reviewer makes the following suggestions:

“I would suggest that much of the general introductory material in paragraphs 1 and 2 be condensed. Also, the last paragraph that summarises the results can be replaced by a single sentence, e.g. ‘To address the role of Bub1 kinase activity we characterised two novel inhibitors BAY-320 and BAY-524’. Condensing this text will then allow the authors to provide more detail with regards to what is known about Bub1. For example, in the Results they describe Sgo1, that H2A is a substrate, the link with Haspin etc.; this should all be in the Introduction. Also, what is the prior art regarding Bub1's kinase activity? They say it is controversial but I don't think that is sufficient. Indeed, there is literature that supports the authors’ claims and this should be included. By way of an example, Perera at al deleted Bub1 in MEFs and added back kinase-dead mutants showing that a catalytic mutant or a mutant lacking the entire kinase domain restored BubR1 localisation, SAC function and chromosome alignment but failed to restore Sgo1 localisation. Klebig et al. described similar findings using an RNAi-complementation assay. Similarly, McGuinness et al. showed that in meiosis I, a Bub1 mutant lacking the entire kinase domain restored SAC function. There are other primary papers that should be cited here, e.g. describing work in various model systems. NB, as alluded to above, the authors could highlight the technical challenges with complementation assays as a means of justifying the use of small molecule inhibitors.”

---

## [Author Response]

Essential revisions:

All three reviewers recognised the quality of the study, and that specific Bub1 kinase inhibitors would be very useful research tools. There were two major experimental concerns, however:

1) More evidence for the specificity of the compounds should be presented. At present this mostly consists of a patent and kinase assays on a panel of kinases.

In the original manuscript we had reported the results of inhibition assays on a panel of 222 protein kinases. To further document the specificity and selectivity of the Bub1 inhibitors we have now performed two additional experiments:

Firstly, using BAY-320 we have conducted an inhibitor-binding assay (DiscoveRx assay) on a vast panel of human kinases, including mitotic kinases. As we show in the new [Supplementary-material SD3-data], BAY-320 showed exquisite binding to Bub1 and at most marginal interaction with any one of 407 other human kinases. This survey thus demonstrates high selectivity of BAY-320 for its target.

Secondly, we have generated dose-response curves using an “in-cell western assay” detecting phospho-Thr 120 signals in mitotically arrested HeLa cells (see new Figure 1—figure supplement 1). This assay yielded an IC_50_ value of 379 +/- 156 nM for BAY-320 (at cellular levels of ATP), in reasonable agreement with the IC_50_ determined by in vitro phosphorylation at 2mM ATP (680 nM).

As with all small molecules, definitive proof of selectivity in a cellular context is nearly impossible to obtain. However, we emphasize that both BAY-320 and BAY-524 have gone through extensive validation in the course of drug development at Bayer; this, combined with the absence of any unpredicted phenotypic effects, makes us highly confident of the selectivity of these compounds.

2) Are the effects of the compounds in cells specific to inhibiting Bub1? The authors are using a relatively high concentration of the inhibitors (3 and 7 μM for BAY-320 and BAY-524, respectively), and the toxic concentration is 10 μM. Moreover, there is still some residual H2A-T120 phosphorylation (Figure 1), which suggests Bub1 activity is not maximally inhibited under these conditions.

Please address these concerns by providing further experimental data. For example, what are the phenotypes of cells treated with the inhibitors in which Bub1 has been depleted and rescued with a kinase-dead construct? If the inhibitors are specific to Bub1 there should be no further effect of the inhibitors.

As we stated above (see response to concern No. 1), dose-response curves obtained from “in-cell western” assays show maximal reduction of Thr120 phosphorylation between 3.0 and 10.0 μM, a concentration range used in key experiments throughout this study. These new results are in full agreement with the data shown in Figure 1, even though they were obtained using different methodology and a different anti-phospho-Thr120 antibody. Thus, although the argument of ‘residual activity’ is difficult to rigorously exclude, we are confident that an increase in concentration >10 μM would be unlikely to change the outcome of our experiments.

To further document the specificity of the phenotypes produced by BAY-B320 and BAY-B524 treatment, we have included additional data to complement the results shown in Figure 1 (see new Figure—figure supplement 1 A and B). Specifically, we compared the effects of inhibitor treatment on Thr120 H2A phosphorylation and Aurora B signals at centromeres/KTs over a vast range of inhibitor concentrations. Interestingly, changes in H2A phosphorylation and Aurora B centromere/KT levels correlate nicely at all concentrations below 10 μM, but no further effect on Aurora B could be observed at concentrations above 10 μM. This argues that, once Bub1 is maximally inhibited (at 10 μM), no further impact on Aurora B localization is triggered by our compounds, attesting to specificity.

From a more general perspective, we emphasize that the two Bub1 inhibitors did not trigger any phenotypes that could not also be observed in response to Bub1 depletion, arguing strongly against off-target effects. Thus, we are confident that our observations specifically reflect Bub1 inhibition.

We have also carefully considered the suggested rescue experiment, but in the end decided against attempting it. Although such an experiment appears plausible at first glance, it would be technically very difficult and, in our opinion, not necessarily conclusive. Any additional effect produced by Bub1 inhibition in a cell depleted of wild-type Bub1 (and rescued by catalytically inactive Bub1) could in fact be attributed to inhibition of residual, depletion-resistant wild-type Bub1.

In addition to this experimental concern, the reviewers suggest some re-writing is required to clarify what is currently known about the function of Bub1 kinase activity. One reviewer makes the following suggestions:“I would suggest that much of the general introductory material in paragraphs 1 and 2 be condensed. Also, the last paragraph that summarises the results can be replaced by a single sentence, e.g. ‘To address the role of Bub1 kinase activity we characterised two novel inhibitors BAY-320 and BAY-524’. Condensing this text will then allow the authors to provide more detail with regards to what is known about Bub1. For example, in the Results they describe Sgo1, that H2A is a substrate, the link with Haspin etc.; this should all be in the Introduction. Also, what is the prior art regarding Bub1's kinase activity? They say it is controversial but I don't think that is sufficient. Indeed, there is literature that supports the authors’ claims and this should be included. By way of an example, Perera at al deleted Bub1 in MEFs and added back kinase-dead mutants showing that a catalytic mutant or a mutant lacking the entire kinase domain restored BubR1 localisation, SAC function and chromosome alignment but failed to restore Sgo1 localisation. Klebig et al. described similar findings using an RNAi-complementation assay. Similarly, McGuinness et al. showed that in meiosis I, a Bub1 mutant lacking the entire kinase domain restored SAC function. There are other primary papers that should be cited here, e.g. describing work in various model systems.”

We have re-written major parts of the Introduction and Results sections. In particular, we have shortened the opening paragraphs of the Introduction, moved introductory material from the former Results section, and discussed prior studies on the role of Bub1 activity in more detail. We are grateful for these referee suggestions and feel that the text of our revised manuscript has gained in clarity.

“NB, as alluded to above, the authors could highlight the technical challenges with complementation assays as a means of justifying the use of small molecule inhibitors.”

To address this comment, we have added a sentence on technical challenges of genetic studies and advantages of small molecule inhibitors (Discussion).